**Brief Communication**

# Microsecond time-resolved X-ray scattering by utilizing MHz repetition rate at second-generation XFELs

Patrick E. Konold[1,11], Leonardo Monrroy [2,11], Alfredo Bellisario [1], Diogo Filipe[1], Patrick Adams[3], Roberto Alvarez[4], Richard Bean [5], Johan Bielecki [5], Szabolcs Bódizs[2,6], Gabriel Ducrocq[7,8], Helmut Grubmueller[7], Richard A. Kirian [5], Marco Kloos[6], Jayanath C. P. Koliyadu [6], Faisal H. M. Koua [5], Taru Larkiala[2,6], Romain Letrun [5], Fredrik Lindsten[7,8], Michael Maihöfer[9], Andrew V. Martin [3], Petra Mészáros[2], Jennifer Mutisya[2], Amke Nimmrich [6,10], Kenta Okamoto [1], Adam Round [5], Tokushi Sato [5], Joana Valerio[5], Daniel Westphal[1], August Wollter[1], Tej Varma Yenupuri[1], Tong You[1], Filipe Maia [1] ✉ & Sebastian Westenhoff [1,6] ✉

Detecting microsecond structural perturbations in biomolecules has wide relevance in biology, chemistry and medicine. Here we show how MHz repetition rates at X-ray free-electron lasers can be used to produce microsecond time-series of protein scattering with exceptionally low noise levels of 0.001%. We demonstrate the approach by examining Jα helix unfolding of a light-oxygen-voltage photosensory domain. This time-resolved acquisition strategy is easy to implement and widely applicable for direct observation of structural dynamics of many biochemical processes.

Biomolecular transformations, reactions and interactions are at the basis of all life. Deciphering these mechanisms in a time-resolved manner and with submolecular precision opens new dimensions of biological understanding. Access to submillisecond timescales in near-native environments is particularly important, but remains challenging.

There are two primary acquisition schemes to acquire time-resolved data. In 'pump-probe' mode, each reaction trigger is followed by a probe pulse at a defined time delay and time-series are constructed by repeated measurement of many time points. This mode enables femtosecond time resolution and has been used at X-ray free-electron lasers (XFELs) for time-resolved protein crystallography and protein solution scattering[1–3]. In practice, this method limits

acquisition rates leading to larger sample consumption. An alternative approach is to read out a series of probe pulses following a single trigger event. In this way, the efficiency of data collection is vastly improved, reducing sample consumption and suppressing experimental noise through massive averaging[4]. Here, the time resolution is limited by the X-ray repetition and detector acquisition rates.

MHz repetition rates at second-generation XFELs now open up the opportunity to use the latter scheme for time-resolved studies in the microsecond range. The European XFEL (EuXFEL) is the first in this class and delivers trains at 10 Hz containing up to 2,700 X-ray pulses with a variable repetition rate up to 4.5 MHz (Fig. 1b)[5]. Thus far, the high repetition rate has posed severe technical challenges for single-pulse

[1]Laboratory of Molecular Biophysics, Department of Cell and Molecular Biology, Uppsala University, Uppsala, Sweden. [2]Department of Chemistry - BMC, Uppsala University, Uppsala, Sweden. [3]School of Science, STEM College, RMIT University, Melbourne, Victoria, Australia. [4]Department of Physics, Arizona State University, Tempe, AZ, USA. [5]European XFEL, Schenefeld, Germany. [6]Department of Chemistry and Molecular Biology, University of Gothenburg, Gothenburg, Sweden. [7]Department of Computer and Information Science (IDA), Linköping University, Linköping, Sweden. [8]The Division of Statistics and Machine Learning (STIMA), Linköping University, Linköping, Sweden. [9]Max Planck Institute for Multidisciplinary Sciences, Göttingen, Germany. [10]Department of Chemistry, University of Washington, Seattle, WA, USA. [11]These authors contributed equally: Patrick E. Konold, Leonardo Monrroy. ✉e-mail: filipe.maia@icm.uu.se; sebastian.westenhoff@kemi.uu.se

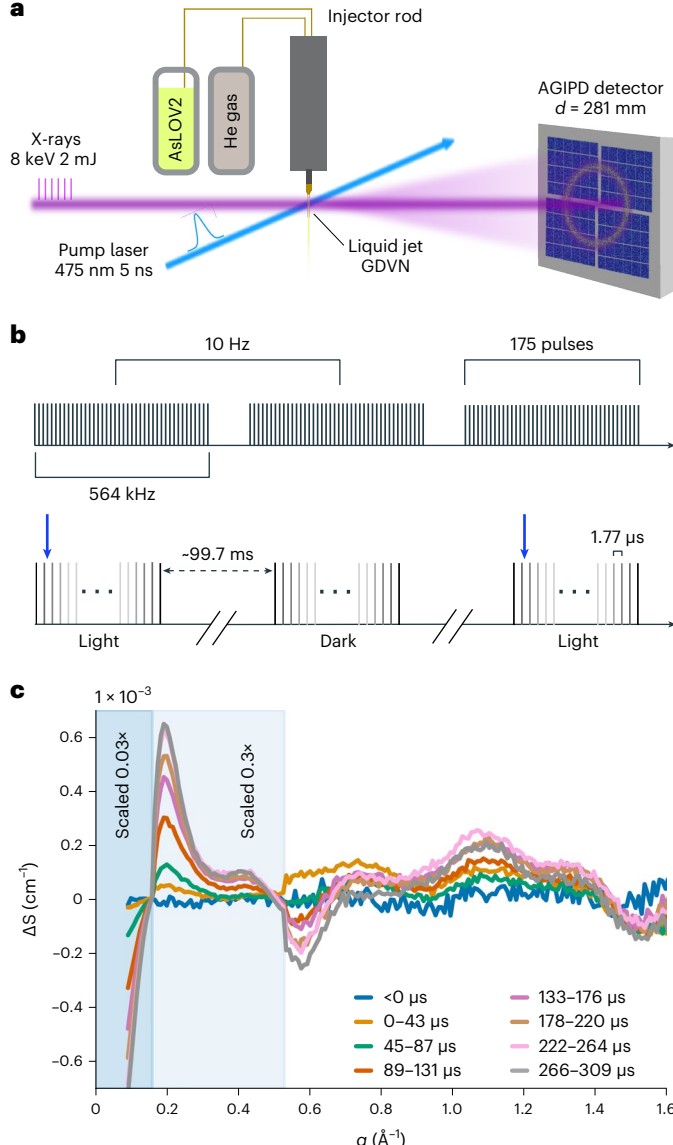

**Fig. 1 | Microsecond TR-WAXS utilizing the MHz repetition rate at the EuXFEL.**
**a**, Schematic depiction of the X-ray and optical laser path, GDVN liquid jet and recorded scattering with the AGIPD detector (not drawn to scale). **b**, Pulse train structure and laser excitation scheme used to obtain microsecond time resolution. The 10 Hz trains comprise 175 pulses at 564 kHz (1.77-μs interval). The blue arrow depicts the timing of optical excitation of every other pulse train. **c**, TR-WAXS data of AsLOV2. The momentum transfer is defined as $q = 4\pi \sin\theta/\lambda$, with $2\theta$ and $\lambda$ as the scattering angle and the X-ray wavelength, respectively. The data were normalized in the $q$-range 1.6 Å$^{-1}$ > $q$ > 1.4 Å$^{-1}$ and scaled for better visualization as indicated in the panel.

detection of scattering and diffraction images, due to electronic noise and nonlinear gain in the detector readout, as well as shockwaves or explosions in the jet[6]. For these reasons, this unique timing capability has only been used in X-ray microscopy, dynamic compression experiments and X-ray photon correlation spectroscopy[7–9], but not yet in the pursuit of biomolecular structural dynamics through protein scattering. Here, we demonstrate the realization of this approach through time-resolved wide-angle X-ray scattering (TR-WAXS) at the EuXFEL.

TR-WAXS can resolve structural changes of biomolecules and chemicals in solution, providing an ~atomic-scale glimpse of their function under near-native conditions[4,10,11]. We investigate the phototropin light-oxygen-voltage (LOV)2 domain from *Avena sativa* (AsLOV2), which features a prototypical signaling mechanism, where a C-terminal

helix (Jα, 22 residues; Extended Data Fig. 1) detaches from the core in response to photoexcitation[12,13]. This unique photoactivity has been exploited in a broad range of optogenetic applications and has been the subject of intense experimental investigation[14–19]. Despite this interest, the mechanism and timing of Jα unfolding and the structure of the unfolded state are not definitively known.

To record microsecond TR-WAXS at the EuXFEL, the sample was carried in a liquid jet via three-dimensional (3D)-printed gas dynamic virtual nozzle (GDVN)[20] to the interaction point of the optical and X-ray beams at the single particles, clusters, and biomolecules and serial femtosecond crystallography (SPB/SFX) endstation (Fig. 1a)[21]. Photoexcitation was conducted through the transparent GDVN nozzle with nanosecond laser pulses timed to the start of every second X-ray pulse train (Fig. 1b). Careful consideration was given to ensure sufficient excitation volume to span the entire X-ray probe train (~1 nl). The scattering was recorded on the AGIPD detector for each probe pulse, covering a $q$-range from 2.1 Å$^{-1}$ > $q$ > 0.08 Å$^{-1}$ (corner resolution). The two-dimensional (2D) scattering was integrated into rings as a function of the momentum transfer ($q$) and delay time ($t$) along the pulse train. Approximately 30% of the data were excluded, because the shape of the scattering was affected by fluctuations in experimental conditions (Methods). After averaging over several repeats, the difference scattering $\Delta S = S_{\text{light}}(q,t) - S_{\text{dark}}(q,t)$ was computed (Fig. 1c). We found that it was crucial to subtract entire laser-on from the laser-off trains from each other, reducing the effect of systematic noise from the detector and fluctuations in jet thickness and X-ray intensity (Extended Data Fig. 2). This reduction was effective as the noise of the $\Delta S$ signal (Extended Data Fig. 3) was comparable to the estimated Poisson noise (Extended Data Fig. 4). The experimental time resolution of 1.77 μs corresponds to the inverse of the repetition rate of the XFEL (564 kHz) and the data span a time window of ~300 μs reflecting the length of the X-ray pulse train.

The TR-WAXS response of AsLOV2 shows microsecond evolution with oscillations extending beyond $q$-values of 1.5 Å$^{-1}$, which translates into a spatial resolution of 4.2 Å. The data have an exceptionally low noise floor corresponding to 0.001% as determined from noise fluctuations from a pre-excitation time point (Extended Data Fig. 3), which is at least one order of magnitude lower than previous accounts for this method[22]. Deconvolution of the data using spectral decomposition with exponential conversion laws indicated that the data are best fit to a sequential model of type A → B → C, yielding base patterns for the three states (Fig. 2b, Extended Data Fig. 5). In TR-WAXS, large difference signals at low $q$ < 0.15 Å$^{-1}$ typically indicate changes of the radius of gyration ($Rg$) of the protein[3]. From this we deduce that the structural change in state C is sizable, but that changes in states A and B are comparably smaller. We assign state C to the unfolded state (vide infra), which is further underpinned by its timescale, emerging within ~300 μs (Fig. 2a), in agreement with kinetics inferred from infrared spectroscopy[17,18]. States A and B could only be resolved because of the low noise floor of the new scattering method approach. State A forms within the first time point of our measurement at 1.77 μs, in agreement with previous reports of FMN-cysteinyl adduct formation[23]. We assign state B to a previously unrecognized intermediate state, which occurs subsequent to Cys adduct formation and before large changes in the Jα helix. Notably, intermediate states in Jα unfolding have been previously proposed through a long MD simulation[24], but not clearly observed experimentally.

Focusing on state C and to assess the extent of Jα unfolding, we refined structural models predicted by AlphaFold[25], where a large variability was obtained through sampling with dropout enabled inference (20,000 structures predicted)[26] and a number of glycine mutations in the Jα helix. We then determined best fits against the predicted structures by comparison of the root-mean-square of residuals ($R^2$) between theoretical and experimental difference scattering curves (Fig. 2d). Since we compare the curves on absolute scales, this selection is also

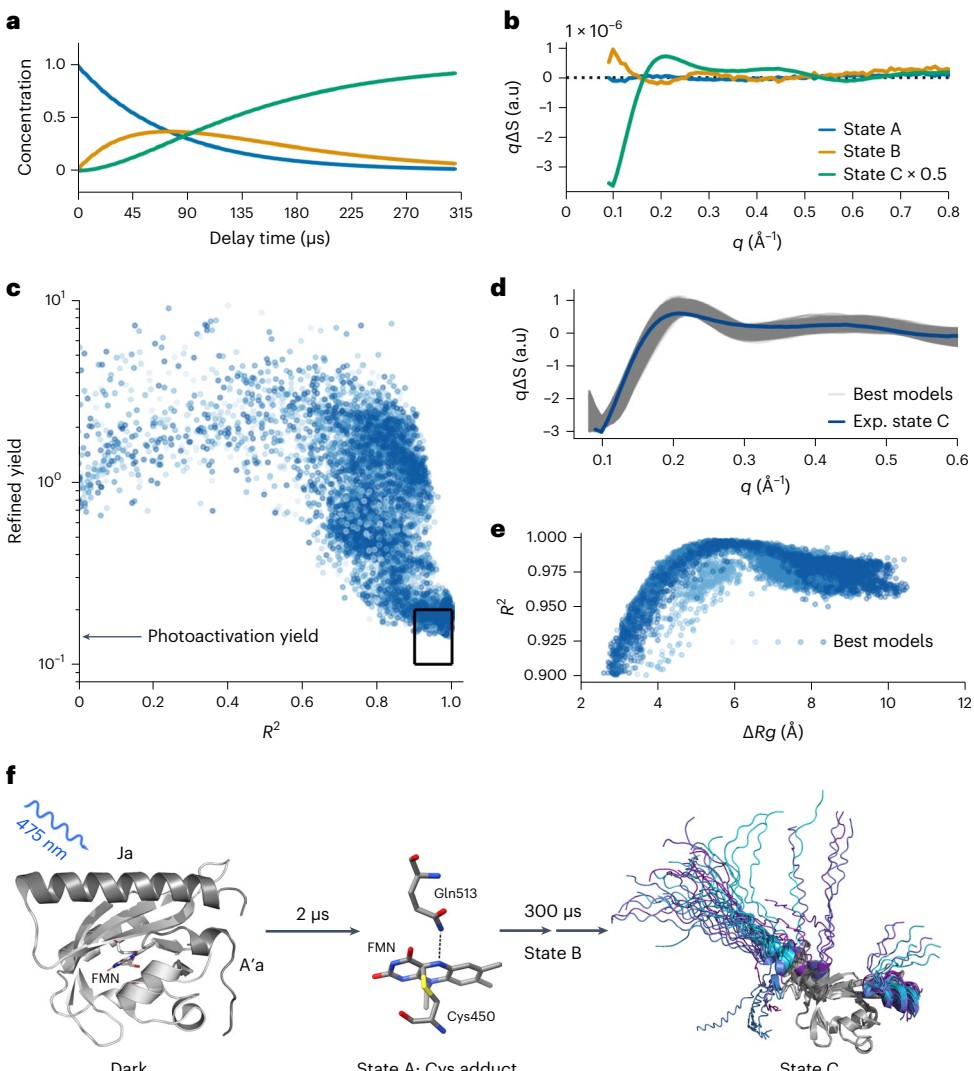

**Fig. 2 | TR-WAXS uncovers a new intermediate state and the structure of the unfolded Jα helix in the AsLOV2 photoreceptor domain. a,b,** The time evolution of constituent states (**a**) and their spectral components derived from kinetic decomposition of the TR-WAXS data (**b**). **c,** Structural modeling results generated using our adapted AlphaFold method. $R^2$ is used as an indicator of a good fit between experimental and theoretical difference signals. Darker blue shades correspond to increasing numbers of mutations in the Jα helix. Structures with mutations in the N-terminal helix are also included. The best models were selected by choosing those that have both a photoactivation yield of 15 ± 5% (as derived in Extended Data Fig. 9) and $R^2 > 0.9$, resulting in 6,032 candidate models (black box). **d,** The theoretical difference scattering of the best fits (gray) and the scaled experimental scattering profile of state C (blue) are shown. **e,** $R^2$ of the top candidate structures versus change in radius of gyration ($\Delta R$g). **f,** The structural dynamics results are shown in the canonical photoactivation mechanism of AsLOV2.

based on appropriate computed activation factors of the structural pairs (further described in the Methods, the boxed region in Fig. 2c includes 6,032 structures). All of the selected structures show unfolded Jα helices (subset shown in Fig. 2f), with an increase of $R$g by 5–7 Å yielding the best fits (Fig. 2e). Notably, an inspection of the best-fitting models shows that the residues directly preceding the Jα segment, which form a loop segment in the dark, now form an ordered helical domain (Extended Data Fig. 6). Finally, we find that the N-terminal A'α helix is unfolded in most structures. Our data establishes that the Jα helix unfolds in a two-step mechanism within 300 µs, and also suggests that it completely unfolds and that additional structural changes accompany this process. This concludes a long series of investigations into Jα unfolding[14–19,24], and demonstrates the promising capability of this new time-resolved X-ray scattering method.

Our new implementation of TR-WAXS realizes the unused potential of MHz XFELs to provide unique structural information about transient states on the important microsecond timescale. The additional timing information is gained with only minor adjustments of existing XFEL acquisition schemes and is highly compatible with other methods that use short X-ray pulses, for example serial crystallography[1–3] or X-ray emission spectroscopy[27]. The method rests on the high X-ray fluence per pulse at the EuXFEL, which is about three orders of magnitude higher than a fourth-generation synchrotron (Extended Data Fig. 7). Paired with the fast readout rate of the AGIPD detector, exceptionally low noise levels are obtained. Currently, this is a unique advantage of second-generation high repetition rate XFELs; however, advances in detector technologies may make synchrotrons competitive in the future. The excellent data quality enabled identification of a new transient state in AsLOV2 Jα unfolding and opens the door for investigating microsecond reaction dynamics with dilute samples of proteins, peptides, RNA or DNA[28], especially when combined with ongoing development of ultrastable liquid sheet jet sample injection technology[29]. It also permits detection of difference scattering signals to very high scattering angles ($q > 1.5$ Å$^{-1}$; Fig. 1c), suggesting that time-resolved

and high-resolution structural information can be obtained in crystallography[30–32] or single-particle diffraction experiments[33]. Overall, we anticipate that our method will accelerate knowledge gain for dynamic enzymatic and chemical mechanisms.

## Online content

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

## Methods

### Sample delivery at EuXFEL SPB/SFX

The sample was delivered to the X-ray interaction region using a liquid jet generated from a 3D-printed type C GDVN provided by the EuXFEL Sample Environment Group. The nozzles used are part of the standard 3D-printed suite offered by the EuXFEL[34]. Fused silica capillaries (0.360 mm outer diameter (o.d.) and 0.150 mm internal diameter (i.d.), Polymicro) were fastened to the liquid and gas inlets using a small dab of epoxy glue (Devcon). The standard SPB/SFX liquid injector rod was used to mount the GDVNs. The rod was assembled as follows: a 3.175 mm o.d. stainless steel tube was first glued to the capillaries ~5 mm above the nozzle. This tube was fastened into a stainless-steel nozzle adaptor using a 10–32 PEEK fitting (Idex). The capillaries were then fed through the entire length of the rod and the end piece was screwed into the tip.

Liquid and gas were delivered to the nozzle as previously described[34]. In short, liquid reservoirs were connected to the nozzle inlets via PEEK tubing (Idex, 0.250 μm i.d.). Multiple reservoirs were connected in parallel to facilitate fast switching between sample, buffer and wash solutions using a high-speed electronic valve (Rheodyne). Liquid flow was regulated using an HPLC pump (Shimadzu LC-20AD), while helium gas flow was regulated with an electronic pressure regulator (Proportion Air GP1). Gas and liquid flow rates of 23 mg min$^{-1}$ and 30 μl min$^{-1}$, respectively were typical during the measurement and monitored with in-line flow meters (Bronkhorst F-111B-2K0-TGD-33-V, 0–700 mg min$^{-1}$ and Bronkhorst ML120V00-TGD-CC-0-S, 0–100 μl min$^{-1}$ respectively). This resulted in jet velocities on the order of 10 s of meters per second, which is sufficient to outrun the radiation-induced explosion caused by the ultrafast X-ray pulse and to replenish the sample for each X-ray exposure.

Alignment of the nozzle tip with respect to the interaction region was carried out by manipulating the position of the injector rod using motorized stages. This placement was aided by visualization with the side-view microscope camera illuminated with the EuXFEL femtosecond laser coupled into the sample chamber via a fiber bundle laser synchronized with the X-ray pulse[35,36].

### Optical excitation scheme

Actinic excitation of the sample was carried out with an optical parametric oscillator (Opolette 355, Opotek) tuned to 475 nm with pulse duration of ~5 ns and pulse energy of 2 mJ mm$^{-2}$ (ref. 35). The output beam (~325 × 338 μm FWHM) was aligned to overlap with the lower half of the GDVN to facilitate excitation of a sufficient sample volume to span the entire X-ray pulse train (~1 nl). Given the slower fluid velocity within the inner GDVN channel, it is possible to excite sufficient volume within the ~325 × 325 μm focal spot. Careful evaluation of the illumination volume was carried out to ensure sufficient sample excitation. The optical pump laser was modulated at half the XFEL intertrain repetition rate (5 Hz) yielding alternating light and dark trains to enable robust extraction of the light-induced scattering response. In this way, the experimental time range and resolution were directly defined by the XFEL pulse bunch length and intra-train repetition rate (~300 μs and 1.77 μs, respectively). This strategy represents a convenient means to access dense temporal sampling on the micro- to millisecond scale that does not require complex electronic triggering or changes to optical beam alignment.

### EuXFEL SPB/SFX beamline configuration

The data were collected at the SPB/SFX instrument of the EuXFEL in September 2022, under the proposal p3046. The EuXFEL delivered bunch trains at 10 Hz with an intra-train pulse repetition rate of 564 kHz. The photon energy was 8,000 eV, which corresponds to a wavelength of 1.55 Å. From previous measurements, the focal spot was estimated at around 300 × 300 nm FWHM. The energy of every X-ray pulse was measured by a gas monitor detector upstream and was close to 2 mJ. With this beamline configuration and photon energy the beamline

transmission between the gas monitor detector and the interaction region is estimated to be 65%. The AGIPD 1 M detector was placed 0.281 m downstream from the interaction region[37]. The experiment was monitored online with Hummingbird[38].

### Data acquisition and computation of time-resolved difference X-ray scattering

We recorded 175 images per pulse train with a time spacing of 1.77 μs between each acquisition. For practical reasons, the data collection was split up into runs, where each run comprised a few thousand trains. Data were collected at two different sample concentrations: 15 and 11 mg ml$^{-1}$. Data filtering was performed to account for intermittent liquid jet instability. This was conducted by comparing the correlation between the absolute integrated scattering intensity of individual trains within the run against the train-average for a run; trains below a threshold of 0.99995 were considered low quality and omitted from the averaging of the scattering curves. A total of 7.75 million images were retained or ~70% of usable frames. The averaging of the scattering curves was conducted for the two concentrations separately over repeats and runs of light and dark absolute scattering, resulting in $S_{light}(q,t)$ and $S_{dark}(q,t)$. Here, $t$ is the delay time of the probe pulse with respect to the arrival time of the excitation laser pulse. If not indicated otherwise (Fig. 1c and Extended Data Fig. 3), the filtered data were then normalized over the entire $q$-range (2.1 Å$^{-1}$ > $q$ > 0.08 Å$^{-1}$) by dividing each scattering point by the sum of the total scattering within the selected $q$-range. Once normalized, difference scattering curves were calculated, $\Delta S = S_{light}(q,t) - S_{dark}(q,t)$. Subsequently, the low concentration was scaled to match the high concentration, a small offset was also applied to account for systematic detector errors, and the difference curves ($\Delta S$) of the two concentrations were then merged using a weighted average, where the weights correspond to the number of light frames in each dataset. The data displayed in Fig. 1c were recorded during 8 h of total experiment time and the duration of pure data collection was 3 h 15 min with a total sample consumption of ~75 mg. Such a quantity is accessible for a wide range of biological materials. Furthermore, a reduction might be possible using a flow segmentation scheme as described by Echelmeier et al.[39].

### Kinetic modeling

Kinetic decomposition of the experimental data was performed to better understand the reaction dynamics of the AsLOV2 photocycle. Global fitting was carried out assuming a sequential reaction scheme with a variable number of states. The TR-WAXS scattering data can be expressed as a linear combination of time-independent basis spectra:

$$\Delta I(q,t) = \sum_i (BS_i(q)C_i(t)) \tag{1}$$

Where $\Delta I(q,t)$ is the measured transient intensity, $BS(q)$ are the time-independent basis spectra and $C(t)$ are the time-dependent concentrations of the components $i$. In the fitting procedure, the time-dependence of a two-state (A → B) and three-state (A → B → C) model were expressed as exponential functions as:

$$C_A(t) = \exp(-k_A t), \tag{2}$$

$$C_B = 1 - C_A, \tag{3}$$

and

$$C_A(t) = \exp(-k_A t), \tag{4}$$

$$C_B(t) = \frac{k_A}{(k_B - k_A)} \times \left[\exp(-k_A t) - \exp(-k_B t)\right], \tag{5}$$

$$C_C(t) = 1 - \frac{1}{(k_B - k_A)} \times \left[k_B \exp(k_A t) - k_A \exp(k_B t)\right]. \tag{6}$$

The constants $k_i$ were optimized using simplex minimization of the target function $r$ using 'fminsearch.m' of MATLAB v.2019:

$$r = \left[\sum_{q,t} \Delta I(q,t) - \sum_i (BS_i(q)C_i(k))\right]^2 \quad (7)$$

whereby $BS_i(q)$ was determined on each iteration by the least-square solution of equation (1) using the backslash operator ('mldivide') in MATLAB. The goodness of the fits was judged by plotting the refined kinetics against time-slices of the data in (Extended Data Fig. 4). The three-state model gave a lower $r$ compared to the two-state model and a better agreement in terms of shape of the fit.

## AlphaFold models
To simulate unfolding of the Jα helix, 2,000 initial AlphaFold models were created using AFsample with dropout enabled, to determine whether the aggressive sampling could capture the unfolding[25,26]. The initial 2,000 models did not display any unfolding and showed only small differences in the last two residues of the Jα helix.

A second approach was taken to ensure the models would have the unfolded helix. By substituting every second amino acid in the helix it can be destabilized artificially, making AlphaFold unable to find any similar sequence in its database. Therefore, it classifies it as a disordered loop instead. The input sequence was modified by introducing 3, 5, 7, 9 and 11 glycine mutations starting from the second-to-last residue and substituting every second until the desired number of mutations was reached. These mutated sequences were then used to run AFsample again, generating 2,000 models for each sequence, resulting in a total of 10,000 models. In addition, we also introduced mutation in the N-terminal helix to investigate its possible effects on the scattering, five glycine inserts were introduced on top of the Jα inserts resulting in an additional 10,000 models.

AFsample was run using the following settings: 1,000 models with dropout templates enabled, 500 models with dropout enabled and no templates, each with a maximum of 21 recycles, and 500 models with dropout enabled and no templates, each with a maximum of 9 recycles. The mutated amino acids were then reverted back to the original residue using coot[40]. From the initial 2,000 models, which showed very small variation in structure, one was chosen to represent the native state of the protein.

## Computation of scattering profiles
Theoretical scattering profiles were calculated using Pepsi-SAXS from the AlphaFold models, a sample concentration of 15 mg ml$^{-1}$ was assumed for the theoretical scattering profiles[41]. We used Pepsi-SAXS, because it computes the scattering of the solvation shell from a grid, which we expect to lead to accurate results for partially unfolded proteins and because the software is very efficient in computing the scattering of the candidate compounds. The scattering was computed for 170 points, in the range between 0.08 Å$^{-1}$ < $q$ < 1.5 Å$^{-1}$. The theoretical difference scattering ($\Delta S_{model}$) was calculated by subtracting scattering of a native predicted model from each of the unfolded models.

## Structural fitting
To compare the theoretical and experimental scattering, the experimental scattering was put on an absolute scale by scaling the experimental dark scattering to the theoretical dark scattering equation (8), this scale was then applied to the difference scattering (Extended Data Fig. 8). For comparing the models to the experiment, we optimized a projected photoactivation yield $c$ for each candidate structural pair according to:

$$SS_{resdark} = \sum_q (S_{dark.exp.} \times c_{abs} - S_{dark.theory})^2 \quad (8)$$

$$R^2 = \frac{\sum_q (c\Delta S_{exp.scaled} - \Delta S_{model})^2}{\sum_q \Delta S^2_{exp.scaled}} \quad (9)$$

$c$ corresponds to the photoactivation yield, which a certain structural pair would require the difference X-ray scattering to have for an optimal fit. By performing a least-square optimization on the numerator in equation (9), c could be estimated for each structural pair. The q-scale used for fitting was $q$ < 0.16 Å$^{-1}$. We used it to discriminate good fits. We selected 6,032 models with the highest $R^2$ and which had a projected activation yield of 15 ± 5% (see Supplementary Data).

## Determination of the photoexcitation yield by auxiliary SAXS experiments
To determine the photoexcitation (activation) yield, we performed a separate SAXS experiment with AsLOV2. The data was recorded at the Diamond Light Source (beamline B21) at room temperature (20 °C). First, we recorded SAXS data in complete darkness followed by illumination of the sample for one second using a laser diode (wavelength, 470 nm; average power, 68 W m$^{-2}$; spot size was an ellipse of 4.5 × 1.9 mm with an area of 6.7 mm$^2$). The protein was allowed to dark-revert for 5 min and the procedure was repeated four more times with increasing illumination time for each cycle (2, 5, 10 and 20 s). Saturation of the difference signal (light–dark) was observed from 5 s of illumination onwards (Extended Data Fig. 9b). By comparing the signal height of this SAXS difference data for full photoconversion to $\Delta S_{exp}$ from the XFEL, we determined the excitation yield at the EuXFEL to be 15 ± 5% (Extended Data Fig. 9d).

## Protein expression and purification
AsLOV2 expression and purification followed from previously reported protocol[12]. The expression plasmid (6His-Gb1-AsLOV2) was obtained from the group of K. Gardner at CUNY. The AsLOV2 was expressed in *Escherichia coli* BL21 (DE3) STAR (Thermo Fisher Scientific). This culture was propagated in 11 l of the LB medium, induced with 1 mM IPTG at OD$_{600}$ = 0.8–0.9 and then incubated at 18 °C, 180 rpm for 16 h. The cells were centrifuged at 6,000$g$, 4 °C for 20 min. The cell pellet was washed with 30 ml Tris buffer (50 mM Tris-HCl, 500 mM NaCl, 0.5 mM dithiothreitol and 5% ($v/v$) glycerol, pH 8). The washed pellet was then resuspended in 60 ml Tris buffer and sonicated for 2 min with cycles of 15 s of sonication separated by 45-s intervals at 50% pulse amplitude using a Branson 450 Digital Sonifier (Branson, BBU13119802A). The sonicated lysate was then cleared at 15,000$g$, 4 °C for 35 min and filtered with a 0.45-μm filter. This was then equilibrated within a Ni-NTA resin column (88222, Thermo Scientific) for further purification. The resin was washed with Tris buffer and 50 mM imidazole. Then, elution was performed in steps up to 500 mM imidazole. The 6×His tag was cleaved with TEV protease (T4455, Sigma Aldrich) in a 1:50 molar ratio of TEV:AsLOV2, and the mixture was dialyzed twice in Tris buffer at 4 °C. The dialyzed AsLOV2 sample was applied to another Ni-NTA resin column to remove the cleaved 6×His tag and residual TEV. The AsLOV2 sample was further purified using a HiPrep 26/60 Sephacryl S-100 HR (17119401, Cytiva) size-exclusion column. The final yield was 325 mg (from 11 l culture) and it was stored at −80 °C before the experiment at 13 mg ml$^{-1}$.

## Reporting summary
Further information on research design is available in the Nature Portfolio Reporting Summary linked to this article.

## Data availability
The raw experimental data are available at the EuXFEL repository at https://doi.org/10.22003/XFEL.EU-DATA-003046-00 and the radial profiles are available at the Coherent X-ray Imaging Data Bank, CXIDB ID 225. The refined AlphaFold models are available as Supplementary Data. Source data are provided with this paper.

**Brief Communication** https://doi.org/10.1038/s41592-024-02344-0

## Code availability

The code utilized in this research is solely based on established equations and no new central algorithm was developed or utilized in the process; however, the code can be provided upon request.

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

## Acknowledgements

S.W., F.M. and H.G. acknowledge support by the Swedish Research Council through research award 2019-06092 and the BMBF for this research. F.M. acknowledges further support by the Swedish Research Council through research award 2018-00234. The sample reservoirs employed in parts of the measurements presented here were designed and fabricated by the Max Planck Institute for Medical Research, Heidelberg, which also provided instruction in its use. P.A. and A.V.M. acknowledge the support of the Universities Australia and the German Academic Exchange Service for this research. We thank K. Gardner at CUNY for sharing the AsLOV2 plasmid. We acknowledge the use of the EuXFEL biological sample preparation laboratory, enabled by the XBI User Consortium. We acknowledge the EuXFEL in Schenefeld, Germany for provision of X-ray free-electron laser beamtime at Scientific Instrument SPB/SFX and thank the SPB/SFX instrument group and facility staff for their assistance. J.V. acknowledges funding from the European Union's Horizon 2020 research and innovation program under grant agreement no. 101004728. R.K. acknowledges funding from the National Science Foundation, BioXFEL Science and Technology Center (award no. 1231306) and Directorate for Biological Sciences (award no. 1943448). AlphaFold computations were enabled by resources provided by the National Academic Infrastructure for Supercomputing in Sweden at NSC Berzelius partially funded by the Swedish Research Council through grant agreement no. 2022-06725.

## Author contributions

S.W., F.M. and P.K. designed the study. D.F. prepared the protein samples. L.M., F.M. and A.B. reduced the data. P.K., L.M., A.B., D.F., P.A., R.A., R.B., J.B., S.B., G.D., H.G., R.K., M.K., J.P., F.K., T.L., R.L., F.L., M.M., A.M., P.M., J.M., A.N., K.O., A.R., T.S., J.V., D.W., A.W., T.V.Y., T.Y., F.M. and S.W. collected the scattering data at the EuXFEL. L.M. analyzed the time-resolved scattering data. L.M., S.W., P.K. and F.M. wrote the paper with input from all authors.

## Funding

## Competing interests

The authors declare no competing interests.

## Additional information

**Extended data** is available for this paper at https://doi.org/10.1038/s41592-024-02344-0.

**Correspondence and requests for materials** should be addressed to Filipe Maia or Sebastian Westenhoff.

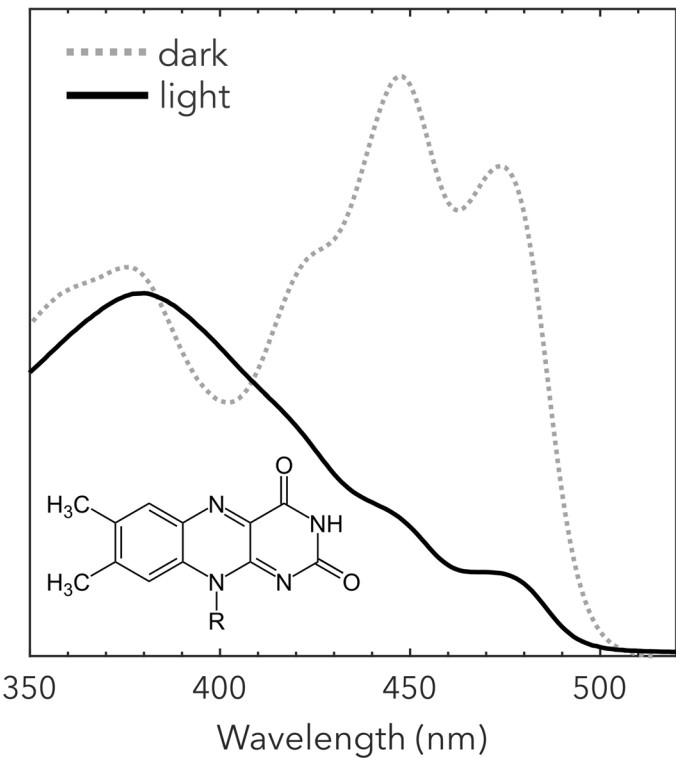

**Extended Data Fig. 1 | UV-VIS absorption spectrum.** AsLOV2 UV-VIS absorption spectrum in dark and light states together with FMN cofactor chemical structure.

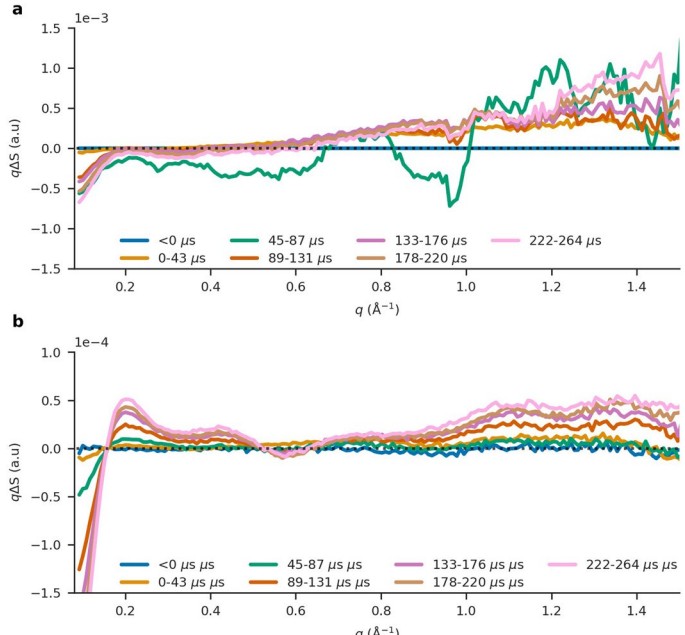

**Extended Data Fig. 2 | Comparison of different ways to compute difference scattering. a**. Here, we present an alternative approach that relies solely on the dark frames recorded during laser-induced trains. The difference scattering is calculated by averaging the dark measurements in each of the light-induced trains and using this average as the dark reference. **b**. Our more reliable approach of calculating the difference scattering is to subtract an entire dark train from a light train. This removes systematic errors from the data acquisition in the detector. Each time point in the plots is composed of 50 pulses binned together, and the averaged dark scattering is subtracted to produce the difference scattering. The difference scattering has been scaled to match the theoretical scattering from Pepsi-SAXS[10].

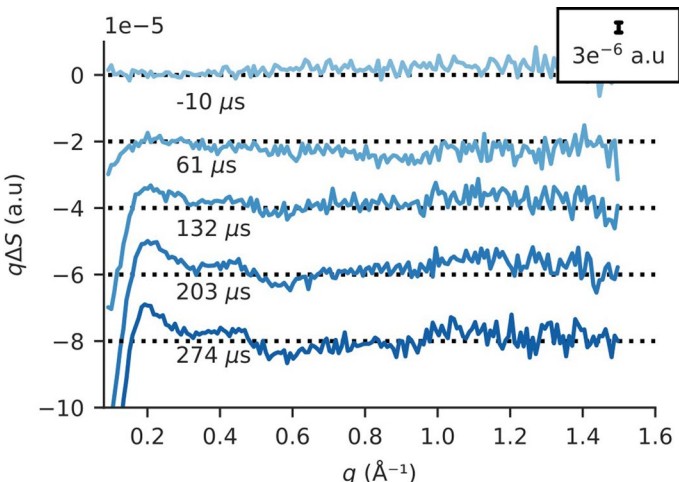

**Extended Data Fig. 3 | Time-resolved WAXS data collected with low noise levels recorded at the SPB/SFX instrument of the European XFEL.** Our data shows an exceptional signal-to-noise ratio, allowing for difference signals of -$10^{-6}$ to be recorded. As the normalization conditions alter the scale of the difference signal, the data is typically normalized around 1.5 Å⁻¹ or 2.3 Å⁻¹. Here, the data is normalized to the scattering in the range between 1.6 Å⁻¹ and 1.4 Å⁻¹ to match the data treatment of previously published time-resolved solution scattering studies performed at a synchrotron, where best-case noise levels are -$10^{-4}$ and -$5*10^{-5}$ in pump-probe and sequential acquisition schemes. The data shown here, $\Delta I(q)$, have a q-bin size of 0.00836 Å⁻¹. The red lines at the pre-excitation time point show $\pm 10^{-5}$, respectively; this boundary corresponds to the noise levels in the data.

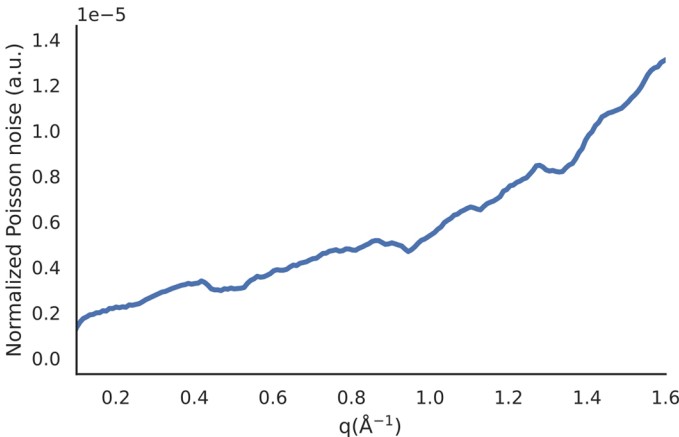

**Extended Data Fig. 4 | Estimated relative poisson noise of the experiment.** The relative poisson noise was estimated from $\sqrt{I(q)}$, where $I(q)$ represents the total number of photons per q-bin for a single time delay, using a bin size of 0.00836 Å$^{-1}$. The data shown here, was normalized to the q-range between 1.6 Å$^{-1}$ and 1.4 Å$^{-1}$. The estimated noise is at the same level as the difference scattering of the pre-excitation time point, where there is no signal by definition, shown in Extended Data Fig. 3, indicating that most noise sources were avoided or eliminated through subtraction when calculating the difference scattering.

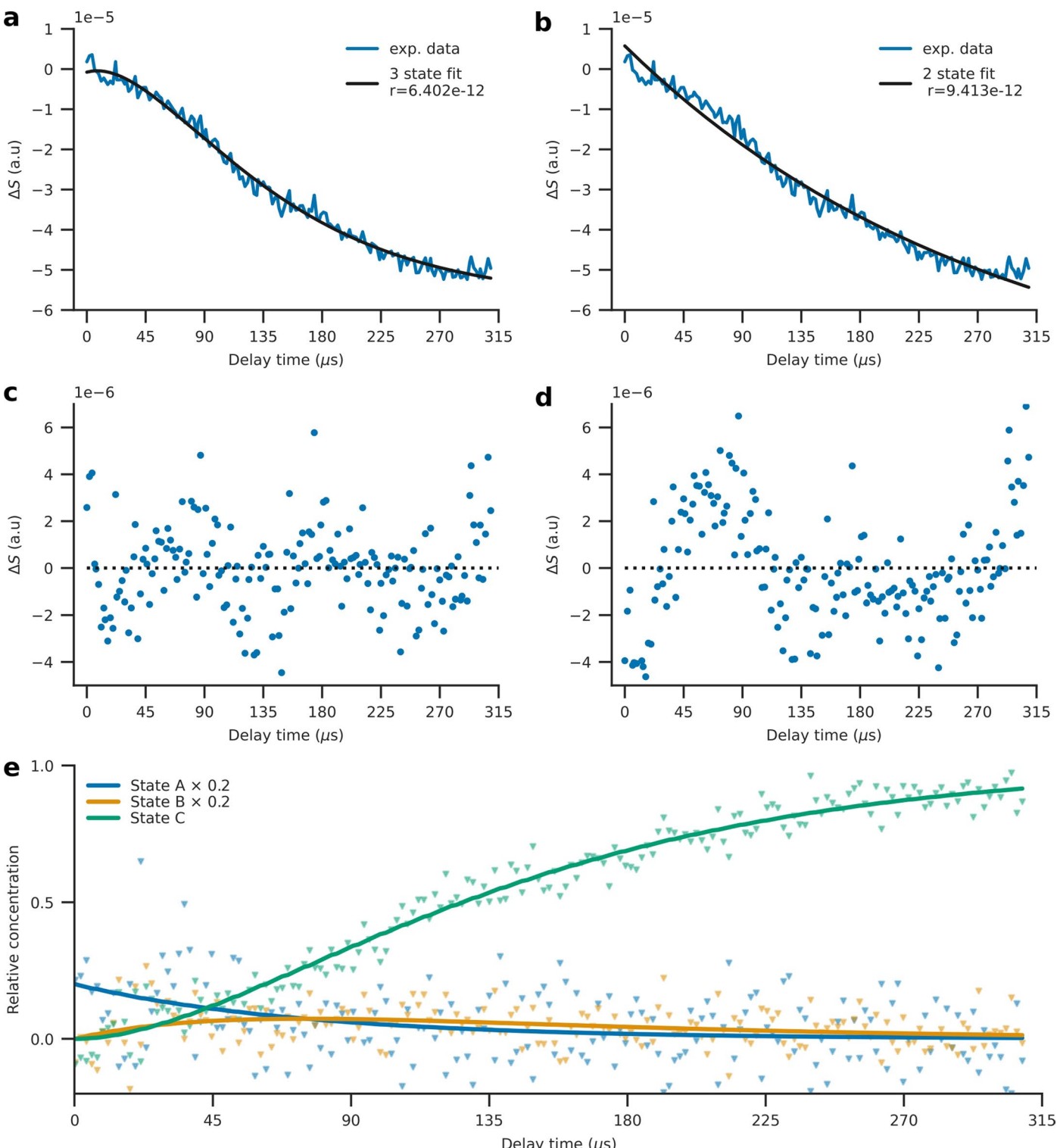

**Extended Data Fig. 5 | A three-state model is required for kinetic fitting of the difference WAXS data.** The reconstructed kinetics and residuals ($r$, see Eq. 7) of the AsLOV2 difference scattering data for a decomposition with a three-state model (**a,c**) are compared to those from a two-state kinetic model (**b, d**). The scattering data and residuals are shown as an average over the q-range (0.08 Å$^{-1}$ < q < 1.2 Å$^{-1}$). The three-state model is preferred over the two-state model, because the fits and residuals of the three-state model show a very good agreement over the entire time-series (panels **a** and **c**), whereas the two-state model shows systematic deviations at early and late times (panels **b** and **d**). Moreover, the $r$ value was better for the three-state model (6.4*10$^{-12}$) compared to the two-state model (9.413*10$^{-12}$). (**e**) The time evolution of the constituent states derived from spectral decomposition are shown including experimental data points. States A and B exhibit high noise because of the low signal amplitude (Fig. 2b), however, as demonstrated, a three-state model is needed to correctly capture the kinetics. States A and B are scaled by a factor of 0.2 for better visualization.

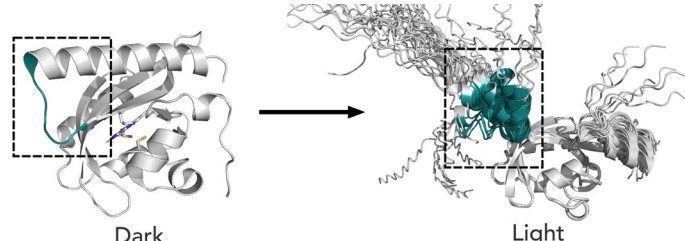

**Extended Data Fig. 6 | Comparison of loop region in folded/unfolded state.** The loop region (shown in green) in the dark structure (PDB 7GPX) converts to a helix in the light following from AlphaFold modeling of State C.

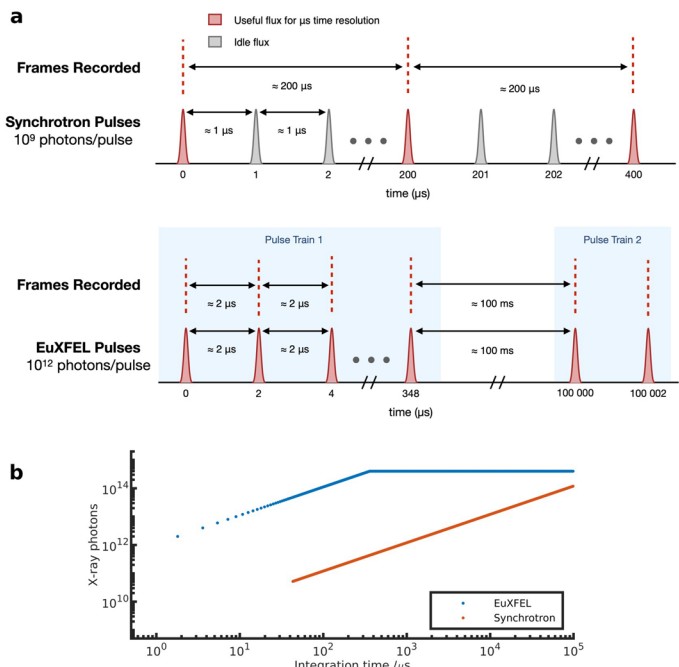

**Extended Data Fig. 7 | Comparison of data acquisition configurations between the European XFEL and a modern synchrotron configuration.**
**(a)** Synchrotrons nowadays can achieve average fluxes approaching the European XFEL. However, the available detectors, such as the Eiger 2, have maximum continuous frame rates on the order of 5000 frames per second. For microsecond time resolution, so detector integration below a microsecond, the usable flux is only about 0.5% of the total. At the European XFEL, the unusual pulse structure makes it possible to use all the available flux. **(b)** Total detectable photon flux at a 4th generation synchrotron versus the EuXFEL source as a function of integration time. The EuXFEL was assumed to have a repetition rate of 564 kHz and $2 \times 10^{12}$ photons/pulse and the synchrotron 1.2 MHz and $1 \times 10^{9}$ photons/pulse. For the synchrotron we assumed an Eiger detector in burst mode at 23k fps, giving a minimum integration time of 43. As timescales increase the performance of a synchrotron approaches that of an XFEL.

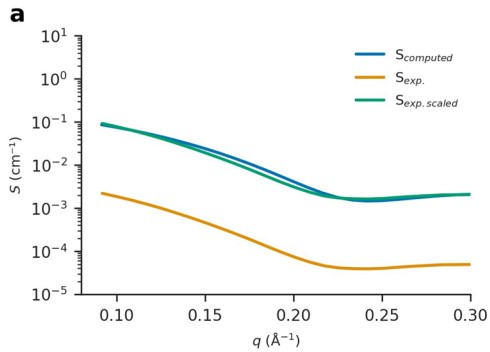

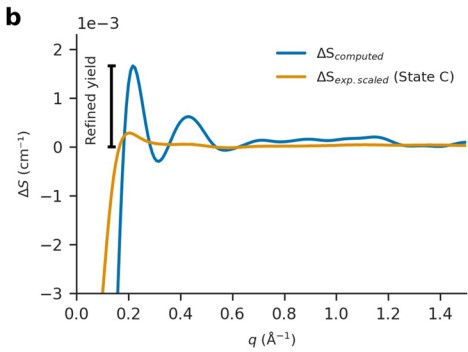

**Extended Data Fig. 8 | Scaling procedure to assign units to the experimental difference X-ray scattering. (a)** The buffer-subtracted and averaged experimental X-ray scattering in dark is scaled to the scattering computed from the dark model of AsLOV2 ($S_{computed}$). **(b)** The difference scattering ($\Delta S_{exp}$) was scaled using the same factor to ensure comparability with the theoretical scattering ($\Delta S_{computed}$). In this context, $\Delta S$computed represents the computed difference scattering between a candidate structure for state C and the ground state structure. Now, the scaling factor between the experimental and theoretical difference scattering, which is shown for one candidate structural pair in the panel, will correspond to the refined photoactivation yield for a candidate structural pair. The factor c was determined by Eq. 8 in the structural fit procedure for each difference scattering curve from candidate structural pairs. The computed scattering was calculated assuming a sample concentration of 15 mg/ml.

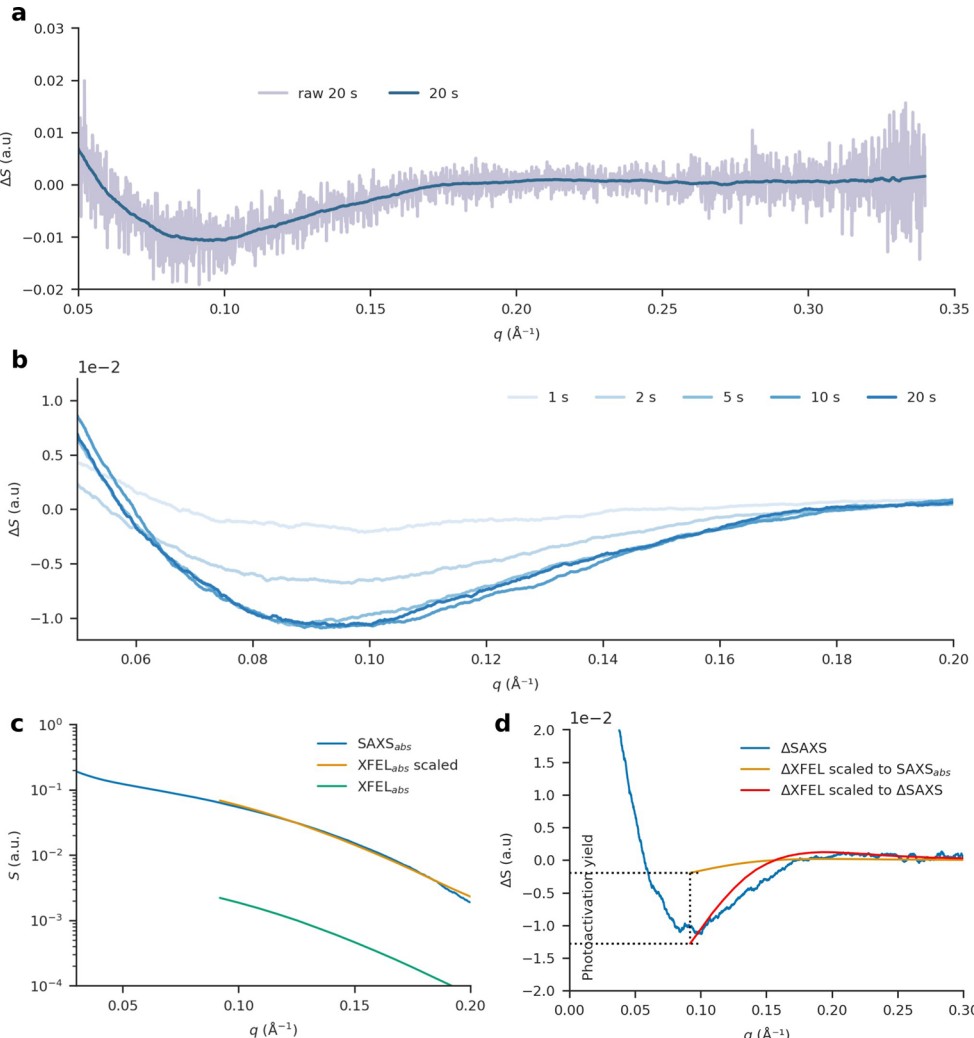

**Extended Data Fig. 9 | Determination of the photoactivation yield from steady state difference SAXS. (a.)** Smoothed difference SAXS curves (light–dark) on top of raw difference SAXS curves for 20 s illumination time. **(b.)** Zoom-in and overlay of the difference SAXS curves (light–dark) at all illumination times of AsLOV2. The difference signal saturated for illuminations of more than five seconds, indicating that the maximum photoactivation yield of 100% had been achieved. **(c.)** The XFEL scattering was then scaled to the SAXS scattering from Diamond to facilitate direct comparison between them. **(d.)** The photoactivation of the XFEL was subsequently determined by comparing the ratio between the scaled XFEL and the saturated SAXS difference scattering. From this analysis, we determined that the photoexcitation yield at the XFEL was approximately 15%. The scattering obtained from the SAXS experiment was smoothed using a Savitzky–Golay filter with a window length of 120 and 1st degree polynomial if nothing else is stated. The SAXS data was recorded at Diamond Light Source, beamline B21.

| | |
|---|---|

# Reporting Summary

## Statistics

For all statistical analyses, confirm that the following items are present in the figure legend, table legend, main text, or Methods section.

| n/a | Confirmed | |
|---|---|---|
| ☐ | ☒ | The exact sample size (*n*) for each experimental group/condition, given as a discrete number and unit of measurement |
| ☐ | ☒ | A statement on whether measurements were taken from distinct samples or whether the same sample was measured repeatedly |
| ☒ | ☐ | The statistical test(s) used AND whether they are one- or two-sided<br>*Only common tests should be described solely by name; describe more complex techniques in the Methods section.* |
| ☒ | ☐ | A description of all covariates tested |
| ☒ | ☐ | A description of any assumptions or corrections, such as tests of normality and adjustment for multiple comparisons |
| ☒ | ☐ | A full description of the statistical parameters including central tendency (e.g. means) or other basic estimates (e.g. regression coefficient) AND variation (e.g. standard deviation) or associated estimates of uncertainty (e.g. confidence intervals) |
| ☒ | ☐ | For null hypothesis testing, the test statistic (e.g. *F*, *t*, *r*) with confidence intervals, effect sizes, degrees of freedom and *P* value noted<br>*Give P values as exact values whenever suitable.* |
| ☒ | ☐ | For Bayesian analysis, information on the choice of priors and Markov chain Monte Carlo settings |
| ☒ | ☐ | For hierarchical and complex designs, identification of the appropriate level for tests and full reporting of outcomes |
| ☐ | ☒ | Estimates of effect sizes (e.g. Cohen's *d*, Pearson's *r*), indicating how they were calculated |

*Our web collection on statistics for biologists contains articles on many of the points above.*

## Software and code

Policy information about availability of computer code

| Data collection | No software |
|---|---|
| Data analysis | Python 3.9.17, MatlabR2022a, AlphaFold2.2.0, PEPSI-SAXS 3.0 |

For manuscripts utilizing custom algorithms or software that are central to the research but not yet described in published literature, software must be made available to editors and reviewers. We strongly encourage code deposition in a community repository (e.g. GitHub). See the Nature Portfolio guidelines for submitting code & software for further information.

## Data

Policy information about availability of data

All manuscripts must include a data availability statement. This statement should provide the following information, where applicable:
- Accession codes, unique identifiers, or web links for publicly available datasets
- A description of any restrictions on data availability
- For clinical datasets or third party data, please ensure that the statement adheres to our policy

The raw experimental data is available at the EuXFEL repository: https://doi.org/10.22003/XFEL.EU-DATA-003046-00 and the radial profiles are available at the Coherent X-ray Imaging Data Bank, CXIDB ID 225: https://cxidb.org/id-225.html. The refined protein structures are available as supplemental data.

## Human research participants

Policy information about studies involving human research participants and Sex and Gender in Research.

| | |
|---|---|
| Reporting on sex and gender | This study did not involve human research participants and gender is irrelevant in this study. |
| Population characteristics | This study did not involve human research participants and gender is irrelevant in this study. |
| Recruitment | This study did not involve human research participants and gender is irrelevant in this study. |
| Ethics oversight | This study did not involve human research participants and gender is irrelevant in this study. |

Note that full information on the approval of the study protocol must also be provided in the manuscript.

# Field-specific reporting

Please select the one below that is the best fit for your research. If you are not sure, read the appropriate sections before making your selection.

☒ Life sciences          ☐ Behavioural & social sciences          ☐ Ecological, evolutionary & environmental sciences

For a reference copy of the document with all sections, see nature.com/documents/nr-reporting-summary-flat.pdf

# Life sciences study design

All studies must disclose on these points even when the disclosure is negative.

| | |
|---|---|
| Sample size | X-ray scattering data was recorded. We recorded 175 images per pulse train with a time spacing of 1.77μs between each acquisition. For practical reasons, the data collection was split up into runs, where each run comprised a few thousands of trains. 7.75 million images were retained (or ~70% of usable frames) after data reduction. |
| Data exclusions | Data was filtered to account for liquid jet instability. This was done by comparing the correlation between individual trains within a run against the train-average of the run. The threshold was set 0.99995 and all trains below the threshold were omitted from further analysis. |
| Replication | The measurement was replicated a thousand of times and the statistical significance of the singals was verified as described in the paper. |
| Randomization | We split our data into three different data sets based on different delay and concentration. Then by calculating the difference scattering for each data set at different time points we could verify the significance of our signal as they all showed the same evolution of the signal through time. |
| Blinding | Blidning was not applicable in this study since S/WAXS data interpretation is based on mathematical models and physical principles rather than subjective judgments. |

# Behavioural & social sciences study design

All studies must disclose on these points even when the disclosure is negative.

| | |
|---|---|
| Study description | Briefly describe the study type including whether data are quantitative, qualitative, or mixed-methods (e.g. qualitative cross-sectional, quantitative experimental, mixed-methods case study). |
| Research sample | State the research sample (e.g. Harvard university undergraduates, villagers in rural India) and provide relevant demographic information (e.g. age, sex) and indicate whether the sample is representative. Provide a rationale for the study sample chosen. For studies involving existing datasets, please describe the dataset and source. |
| Sampling strategy | Describe the sampling procedure (e.g. random, snowball, stratified, convenience). Describe the statistical methods that were used to predetermine sample size OR if no sample-size calculation was performed, describe how sample sizes were chosen and provide a rationale for why these sample sizes are sufficient. For qualitative data, please indicate whether data saturation was considered, and what criteria were used to decide that no further sampling was needed. |
| Data collection | Provide details about the data collection procedure, including the instruments or devices used to record the data (e.g. pen and paper, computer, eye tracker, video or audio equipment) whether anyone was present besides the participant(s) and the researcher, and whether the researcher was blind to experimental condition and/or the study hypothesis during data collection. |
| Timing | Indicate the start and stop dates of data collection. If there is a gap between collection periods, state the dates for each sample cohort. |

| Data exclusions | If no data were excluded from the analyses, state so OR if data were excluded, provide the exact number of exclusions and the rationale behind them, indicating whether exclusion criteria were pre-established. |
|---|---|
| Non-participation | State how many participants dropped out/declined participation and the reason(s) given OR provide response rate OR state that no participants dropped out/declined participation. |
| Randomization | If participants were not allocated into experimental groups, state so OR describe how participants were allocated to groups, and if allocation was not random, describe how covariates were controlled. |

# Ecological, evolutionary & environmental sciences study design

All studies must disclose on these points even when the disclosure is negative.

| Study description | Briefly describe the study. For quantitative data include treatment factors and interactions, design structure (e.g. factorial, nested, hierarchical), nature and number of experimental units and replicates. |
|---|---|
| Research sample | Describe the research sample (e.g. a group of tagged Passer domesticus, all Stenocereus thurberi within Organ Pipe Cactus National Monument), and provide a rationale for the sample choice. When relevant, describe the organism taxa, source, sex, age range and any manipulations. State what population the sample is meant to represent when applicable. For studies involving existing datasets, describe the data and its source. |
| Sampling strategy | Note the sampling procedure. Describe the statistical methods that were used to predetermine sample size OR if no sample-size calculation was performed, describe how sample sizes were chosen and provide a rationale for why these sample sizes are sufficient. |
| Data collection | Describe the data collection procedure, including who recorded the data and how. |
| Timing and spatial scale | Indicate the start and stop dates of data collection, noting the frequency and periodicity of sampling and providing a rationale for these choices. If there is a gap between collection periods, state the dates for each sample cohort. Specify the spatial scale from which the data are taken |
| Data exclusions | If no data were excluded from the analyses, state so OR if data were excluded, describe the exclusions and the rationale behind them, indicating whether exclusion criteria were pre-established. |
| Reproducibility | Describe the measures taken to verify the reproducibility of experimental findings. For each experiment, note whether any attempts to repeat the experiment failed OR state that all attempts to repeat the experiment were successful. |
| Randomization | Describe how samples/organisms/participants were allocated into groups. If allocation was not random, describe how covariates were controlled. If this is not relevant to your study, explain why. |
| Blinding | Describe the extent of blinding used during data acquisition and analysis. If blinding was not possible, describe why OR explain why blinding was not relevant to your study. |

Did the study involve field work?  ☐ Yes  ☐ No

## Field work, collection and transport

| Field conditions | Describe the study conditions for field work, providing relevant parameters (e.g. temperature, rainfall). |
|---|---|
| Location | State the location of the sampling or experiment, providing relevant parameters (e.g. latitude and longitude, elevation, water depth). |
| Access & import/export | Describe the efforts you have made to access habitats and to collect and import/export your samples in a responsible manner and in compliance with local, national and international laws, noting any permits that were obtained (give the name of the issuing authority, the date of issue, and any identifying information). |
| Disturbance | Describe any disturbance caused by the study and how it was minimized. |

# Reporting for specific materials, systems and methods

We require information from authors about some types of materials, experimental systems and methods used in many studies. Here, indicate whether each material, system or method listed is relevant to your study. If you are not sure if a list item applies to your research, read the appropriate section before selecting a response.

## Materials & experimental systems

| n/a | Involved in the study |
|-----|----------------------|
| ☒ ☐ | Antibodies |
| ☒ ☐ | Eukaryotic cell lines |
| ☒ ☐ | Palaeontology and archaeology |
| ☒ ☐ | Animals and other organisms |
| ☒ ☐ | Clinical data |
| ☒ ☐ | Dual use research of concern |

## Methods

| n/a | Involved in the study |
|-----|----------------------|
| ☒ ☐ | ChIP-seq |
| ☒ ☐ | Flow cytometry |
| ☒ ☐ | MRI-based neuroimaging |

# Antibodies

| | |
|---|---|
| Antibodies used | *Describe all antibodies used in the study; as applicable, provide supplier name, catalog number, clone name, and lot number.* |
| Validation | *Describe the validation of each primary antibody for the species and application, noting any validation statements on the manufacturer's website, relevant citations, antibody profiles in online databases, or data provided in the manuscript.* |

# Eukaryotic cell lines

Policy information about cell lines and Sex and Gender in Research

| | |
|---|---|
| Cell line source(s) | *State the source of each cell line used and the sex of all primary cell lines and cells derived from human participants or vertebrate models.* |
| Authentication | *Describe the authentication procedures for each cell line used OR declare that none of the cell lines used were authenticated.* |
| Mycoplasma contamination | *Confirm that all cell lines tested negative for mycoplasma contamination OR describe the results of the testing for mycoplasma contamination OR declare that the cell lines were not tested for mycoplasma contamination.* |
| Commonly misidentified lines (See ICLAC register) | *Name any commonly misidentified cell lines used in the study and provide a rationale for their use.* |

# Palaeontology and Archaeology

| | |
|---|---|
| Specimen provenance | *Provide provenance information for specimens and describe permits that were obtained for the work (including the name of the issuing authority, the date of issue, and any identifying information). Permits should encompass collection and, where applicable, export.* |
| Specimen deposition | *Indicate where the specimens have been deposited to permit free access by other researchers.* |
| Dating methods | *If new dates are provided, describe how they were obtained (e.g. collection, storage, sample pretreatment and measurement), where they were obtained (i.e. lab name), the calibration program and the protocol for quality assurance OR state that no new dates are provided.* |

☐ Tick this box to confirm that the raw and calibrated dates are available in the paper or in Supplementary Information.

| | |
|---|---|
| Ethics oversight | *Identify the organization(s) that approved or provided guidance on the study protocol, OR state that no ethical approval or guidance was required and explain why not.* |

Note that full information on the approval of the study protocol must also be provided in the manuscript.

# Animals and other research organisms

Policy information about studies involving animals; ARRIVE guidelines recommended for reporting animal research, and Sex and Gender in Research

| | |
|---|---|
| Laboratory animals | *For laboratory animals, report species, strain and age OR state that the study did not involve laboratory animals.* |
| Wild animals | *Provide details on animals observed in or captured in the field; report species and age where possible. Describe how animals were caught and transported and what happened to captive animals after the study (if killed, explain why and describe method; if released, say where and when) OR state that the study did not involve wild animals.* |
| Reporting on sex | *Indicate if findings apply to only one sex; describe whether sex was considered in study design, methods used for assigning sex. Provide data disaggregated for sex where this information has been collected in the source data as appropriate; provide overall numbers in this Reporting Summary. Please state if this information has not been collected. Report sex-based analyses where performed, justify reasons for lack of sex-based analysis.* |

| Field-collected samples | *For laboratory work with field-collected samples, describe all relevant parameters such as housing, maintenance, temperature, photoperiod and end-of-experiment protocol OR state that the study did not involve samples collected from the field.* |
| Ethics oversight | *Identify the organization(s) that approved or provided guidance on the study protocol, OR state that no ethical approval or guidance was required and explain why not.* |

Note that full information on the approval of the study protocol must also be provided in the manuscript.

# Clinical data

Policy information about clinical studies

All manuscripts should comply with the ICMJE guidelines for publication of clinical research and a completed CONSORT checklist must be included with all submissions.

| Clinical trial registration | *Provide the trial registration number from ClinicalTrials.gov or an equivalent agency.* |
| Study protocol | *Note where the full trial protocol can be accessed OR if not available, explain why.* |
| Data collection | *Describe the settings and locales of data collection, noting the time periods of recruitment and data collection.* |
| Outcomes | *Describe how you pre-defined primary and secondary outcome measures and how you assessed these measures.* |

# Dual use research of concern

Policy information about dual use research of concern

## Hazards

Could the accidental, deliberate or reckless misuse of agents or technologies generated in the work, or the application of information presented in the manuscript, pose a threat to:

No | Yes
☐ | ☐ Public health
☐ | ☐ National security
☐ | ☐ Crops and/or livestock
☐ | ☐ Ecosystems
☐ | ☐ Any other significant area

## Experiments of concern

Does the work involve any of these experiments of concern:

No | Yes
☐ | ☐ Demonstrate how to render a vaccine ineffective
☐ | ☐ Confer resistance to therapeutically useful antibiotics or antiviral agents
☐ | ☐ Enhance the virulence of a pathogen or render a nonpathogen virulent
☐ | ☐ Increase transmissibility of a pathogen
☐ | ☐ Alter the host range of a pathogen
☐ | ☐ Enable evasion of diagnostic/detection modalities
☐ | ☐ Enable the weaponization of a biological agent or toxin
☐ | ☐ Any other potentially harmful combination of experiments and agents

# ChIP-seq

## Data deposition

☐ Confirm that both raw and final processed data have been deposited in a public database such as GEO.

☐ Confirm that you have deposited or provided access to graph files (e.g. BED files) for the called peaks.

| Data access links
May remain private before publication. | *For "Initial submission" or "Revised version" documents, provide reviewer access links.  For your "Final submission" document, provide a link to the deposited data.* |
| Files in database submission | *Provide a list of all files available in the database submission.* |
| Genome browser session
(e.g. UCSC) | |

*Provide a link to an anonymized genome browser session for "Initial submission" and "Revised version" documents only, to enable peer review. Write "no longer applicable" for "Final submission" documents.*

## Methodology

Replicates
*Describe the experimental replicates, specifying number, type and replicate agreement.*

Sequencing depth
*Describe the sequencing depth for each experiment, providing the total number of reads, uniquely mapped reads, length of reads and whether they were paired- or single-end.*

Antibodies
*Describe the antibodies used for the ChIP-seq experiments; as applicable, provide supplier name, catalog number, clone name, and lot number.*

Peak calling parameters
*Specify the command line program and parameters used for read mapping and peak calling, including the ChIP, control and index files used.*

Data quality
*Describe the methods used to ensure data quality in full detail, including how many peaks are at FDR 5% and above 5-fold enrichment.*

Software
*Describe the software used to collect and analyze the ChIP-seq data. For custom code that has been deposited into a community repository, provide accession details.*

# Flow Cytometry

## Plots

Confirm that:

☐ The axis labels state the marker and fluorochrome used (e.g. CD4-FITC).

☐ The axis scales are clearly visible. Include numbers along axes only for bottom left plot of group (a 'group' is an analysis of identical markers).

☐ All plots are contour plots with outliers or pseudocolor plots.

☐ A numerical value for number of cells or percentage (with statistics) is provided.

## Methodology

Sample preparation
*Describe the sample preparation, detailing the biological source of the cells and any tissue processing steps used.*

Instrument
*Identify the instrument used for data collection, specifying make and model number.*

Software
*Describe the software used to collect and analyze the flow cytometry data. For custom code that has been deposited into a community repository, provide accession details.*

Cell population abundance
*Describe the abundance of the relevant cell populations within post-sort fractions, providing details on the purity of the samples and how it was determined.*

Gating strategy
*Describe the gating strategy used for all relevant experiments, specifying the preliminary FSC/SSC gates of the starting cell population, indicating where boundaries between "positive" and "negative" staining cell populations are defined.*

☐ Tick this box to confirm that a figure exemplifying the gating strategy is provided in the Supplementary Information.

# Magnetic resonance imaging

## Experimental design

Design type
*Indicate task or resting state; event-related or block design.*

Design specifications
*Specify the number of blocks, trials or experimental units per session and/or subject, and specify the length of each trial or block (if trials are blocked) and interval between trials.*

Behavioral performance measures
*State number and/or type of variables recorded (e.g. correct button press, response time) and what statistics were used to establish that the subjects were performing the task as expected (e.g. mean, range, and/or standard deviation across subjects).*

## Acquisition

Imaging type(s)
> *Specify: functional, structural, diffusion, perfusion.*

Field strength
> *Specify in Tesla*

Sequence & imaging parameters
> *Specify the pulse sequence type (gradient echo, spin echo, etc.), imaging type (EPI, spiral, etc.), field of view, matrix size, slice thickness, orientation and TE/TR/flip angle.*

Area of acquisition
> *State whether a whole brain scan was used OR define the area of acquisition, describing how the region was determined.*

Diffusion MRI      ☐ Used          ☐ Not used

## Preprocessing

Preprocessing software
> *Provide detail on software version and revision number and on specific parameters (model/functions, brain extraction, segmentation, smoothing kernel size, etc.).*

Normalization
> *If data were normalized/standardized, describe the approach(es): specify linear or non-linear and define image types used for transformation OR indicate that data were not normalized and explain rationale for lack of normalization.*

Normalization template
> *Describe the template used for normalization/transformation, specifying subject space or group standardized space (e.g. original Talairach, MNI305, ICBM152) OR indicate that the data were not normalized.*

Noise and artifact removal
> *Describe your procedure(s) for artifact and structured noise removal, specifying motion parameters, tissue signals and physiological signals (heart rate, respiration).*

Volume censoring
> *Define your software and/or method and criteria for volume censoring, and state the extent of such censoring.*

## Statistical modeling & inference

Model type and settings
> *Specify type (mass univariate, multivariate, RSA, predictive, etc.) and describe essential details of the model at the first and second levels (e.g. fixed, random or mixed effects; drift or auto-correlation).*

Effect(s) tested
> *Define precise effect in terms of the task or stimulus conditions instead of psychological concepts and indicate whether ANOVA or factorial designs were used.*

Specify type of analysis:     ☐ Whole brain     ☐ ROI-based     ☐ Both

Statistic type for inference
(See Eklund et al. 2016)
> *Specify voxel-wise or cluster-wise and report all relevant parameters for cluster-wise methods.*

Correction
> *Describe the type of correction and how it is obtained for multiple comparisons (e.g. FWE, FDR, permutation or Monte Carlo).*

## Models & analysis

| n/a | Involved in the study |
|-----|-----------------------|
| ☐ | ☐ Functional and/or effective connectivity |
| ☐ | ☐ Graph analysis |
| ☐ | ☐ Multivariate modeling or predictive analysis |

Functional and/or effective connectivity
> *Report the measures of dependence used and the model details (e.g. Pearson correlation, partial correlation, mutual information).*

Graph analysis
> *Report the dependent variable and connectivity measure, specifying weighted graph or binarized graph, subject- or group-level, and the global and/or node summaries used (e.g. clustering coefficient, efficiency, etc.).*

Multivariate modeling and predictive analysis
> *Specify independent variables, features extraction and dimension reduction, model, training and evaluation metrics.*

