## [Peer Review File · Nature Methods]

Peer Review Information

Manuscript Title: Microsecond time-resolved X-ray scattering by utilizing MHz repetition rate at second-generation XFELs

Corresponding author name(s): Filipe Maia, Sebastian Westenhoff

Editorial Notes: None

Reviewer Comments & Decisions:

Decision Letter, initial version:

Dear Professor Westenhoff,

Your Brief Communication entitled "Microsecond time-resolved X-ray scattering by utilizing MHz repetition rate at second-generation XFELs" has now been seen by 3 reviewers, whose comments are attached. While they find your work of potential interest, they have raised serious concerns which in our view are sufficiently important that they preclude publication of the work in Nature Methods, at least in its present form.

As you will see, the reviewers raise concerns about whether the methodology constitutes a truly unique experiment that requires a high repetition rate XFEL. We think that strong data demonstrating this would be essential for a Nature Methods publication.

Should further experimental data allow you to fully address this and the other criticisms, we would be willing to look at a revised manuscript (unless, of course, something similar has by then been accepted at Nature Methods or appeared elsewhere). This includes submission or publication of a portion of this work somewhere else. We hope you understand that until we have read the revised paper in its entirety we cannot promise that it will be sent back for peer-review.

If you are interested in revising this manuscript for submission to Nature Methods in the future, please contact me to discuss your appeal before making any revisions. Otherwise, we hope that you find the reviewers' comments helpful when preparing your paper for submission elsewhere.

If you wish to explore other journals and transfer your manuscript please use our manuscript transfer portal. You will not have to re-supply manuscript metadata and files, unless you wish to make modifications. For more information, please see our manuscript transfer FAQ page.

Sincerely yours,

Allison

Allison Doerr, Ph.D.
Chief Editor
Nature Methods

Reviewers' Comments:

Reviewer #1:

Remarks to the Author:

In this Brief Communications, Konold et al. reports the application of unique high repetition rate European XFEL for the light induced structural changes of the Light-Oxygen-Voltage (LOV) photosensory domain by SAXS. This is a nice demonstration of the European XFEL pulse trains that can also be used for other measurements like diffraction methods. The signal to noise of the difference 2D spectrum the authors reported from this measurement is excellent and demonstrates the promise of this approach. Therefore, I generally support this article to be published in the method-oriented journals like Nature Methods. On one hand, I am a bit surprised that such approach has not been practiced yet.

The data collection time of ~ 3 h with 30 μ L/min sample consumption, therefore, about 5mL of total volume, seems to be durable for many biological systems.

Below are more specific comments;

- The SAXS data analysis is complicated and beyond my expertise to evaluate the confidence level of the outcome and needs to rely on the comments of other reviewers, although I agree that the three-phase model fit is better than the two-phase fit in SI Fig. 4. There seems to have an unexplained oscillation in the middle time range of 90 – 225 microseconds in this plot. Would that be another intermediate state?

- The most uncomfortable part is $\sim 30\%$ elimination of the data that contains artifact. The SAXS difference signal is so subtle and how one can confidently select images with and without artifacts? This was not clearly described in the manuscript.

- What is systematically changing over the data collection time, that makes taking consecutive difference spectra so efficient?

Reviewer #2:

Remarks to the Author:

The authors describe the use of superconductive EuXFEL to measure protein dynamics with microsecond time resolution.

The authors consider the results presented in this very short, not so detailed paper, a "new implementation".

I find the results intriguing but I don't think that the paper deserve publication in Nature Methods. Here are some of my concerns:

* flux *

The authors ends up using 175 pulses per train at 10 Hz that is 1750 pulses/s. Assuming 10^{12} ph/pulse, this means $\sim 2 \times 10^{15}$ photons/s. At a 3rd generation synchrotron, 10^9 ph/pulse(10mA) can be delivered to the sample. In few bunch modes, the repetition rate can be as high as few MHz (4 bunches at ESRF are at ~ 1.2 MHz). This translates to $\sim 1 \times 10^{15}$ photons/s very similar to the number obtained at EuXFEL (especially considering high fraction of outliers and the need of 50% duty cycle due to the need of extensive dark measurement). So from a photon counting perspective the two facilities are quite comparable when looking at microsecond time scale.

* sample consumption *

The reasons of being able to collect data so effectively, relies a lot on the fast jet ... this implies a huge sample consumption. The authors mentions a flow rate of 30uL/min. Unless, I am mistaken, the final protein concentration is not given (and it should) but I guess this is in the 1 mM range. Running for few (let's say 10, including setup) hours at such flow rate requires the use of 30 uL/min * 60 mins/hour * 10 hours = 18 mL of solution. This is several times higher than available with many (most?) samples. As the authors certainly know, most experiments at synchrotrons are performed using few capillaries (with few tens of uL solution) and in some exceptional case using small peristaltic pumps and few ml solutions.

* detector *

The authors don't give enough credit to the real workhorse of the experiment they performed, i.e. the AGIPD detector. This (together with the jet) allows to collect data very efficiently.

All in all, I agree that the data quality looks quite nice but I see this as marginal improvement with a very high upfront cost (getting beamtime, producing protein, handling AGIPD data, etc.). Maybe the data collection will span over few days instead of three house, but I am convinced that a well designed synchrotron experiment will result in the same "scientific output" (in terms of understanding) at a much lower cost.

The paper also sidetrack the attention by discussing the modeling of Light-Oxygen-Voltage (LOV) photosensory domain. I find this section not very useful. I understand the interest (and challenges) of modeling such kinds of data ... but this is just not the right paper. Showing data from more proteins would have been more useful.

I would be willing to accept a paper along the lines of the one submitted by the authors, only if substantial new information can ONLY be gained using high rep-rate XFELs. Else it might just be another time resolved scattering paper but not a new methodological development.

Reviewer #3:

Remarks to the Author:

In this paper, Konold et al reported the realization of micro-second time-resolved X-ray scattering using second-generation XFEL and demonstrated this technique in the study of light-initiated J α helix unfolding of a Light-Oxygen-Voltage (LOV) photosensory domain. Many important biological reactions occur at micro-seconds scales, such as foldings/unfoldings of protein alpha-helices and DNA/RNA duplexes. However, due to the technical difficulty, there are few scattering techniques at such time scales. For this reason, this paper could open an avenue to study micro-seconds reactions using x-ray scattering. Given the wide applications of x-ray scattering technique, I believe this paper will attract broad interests among the readership of the Journal. Thus, I would recommend publishing it after the authors address the following questions and comments.

1. Question on the realization of time-resolved x-ray scattering measurements. Considering the high jet velocity (~ 10 m/s), in 300 micro-seconds the solution illuminated by a laser pulse will travel > 3 mm which is much larger than the x-ray beam size (~ 300 nm). The authors mentioned "An alternative approach is to read out a series of probe pulses following a single trigger event. In this way, the efficiency of data collection is vastly improved, reducing sample consumption and suppressing experimental noise through massive averaging", but failed to disclose the details on how the time-resolution was achieved in their time-resolved x-ray scattering measurements. Was the jet translated for time-resolution or the x-ray beam steered to follow the illuminated liquid? If it has been documented previously in any reference paper, please note it is the authors' responsibility to provide sufficient experimental details within the manuscript for the sake of the integrity of the paper.

2. Figure 2 and related text.

(a) Fig 2 missed label f.

(b) In Fig 2f and text, is Cys adduct the State A? If so, please explicitly label it in the figure and text.

(c) I would suggest labelling Cys450 (and Gln 513), and highlighting the portion of protein that will change, in the ground state structure, to guide the readers eyes.

3. Comments on the following statements:

" Our data establishes that (i) the J α helix unfolds in a two-step mechanism within 300 μ s, (ii) that it completely unfolds, and (iii) that additional structural changes accompany this process. This concludes a long series of investigations into J α unfolding [refs 14–19,24], and demonstrates the promising capability of this new time-resolved X-ray scattering method. "

Since the structures used in the data interpretation were picked from a limited conformation space and may not be the only solution, I would suggest weakening the tone on statements (ii) & (iii). For example:

" Our data establishes that the J α helix unfolds in a two-step mechanism within 300 μ s, and also suggests that it completely unfolds and that additional structural changes accompany this process."

4. Supplementary Figure 5. No $\Delta S_{\text{computed}}$ is in Fig S5a. Is it a typo in (a) caption? In Fig S5b, what is $\Delta S_{\text{computed}}$? Difference between theoretical and experimental x-ray scattering for the protein in dark / ground state? Or is it the theoretical difference of x-ray scattering between the ground state and A state?

5. Supplementary Figure 6. Since the authors collected the SAXS data on AsLOV2 with different

illumination times, I would like to encourage the authors display the full range of SAXS data in Fig S6, because they are closely relevant to the time-resolved measurements, and also it would provide a good comparison between steady state and time-resolved measurements.

Author Rebuttal to Initial comments

Reviewer #1:

Remarks to the Author:

In this Brief Communications, Konold et al. reports the application of unique high repetition rate European XFEL for the light induced structural changes of the Light-Oxygen-Voltage (LOV) photosensory domain by SAXS. This is a nice demonstration of the European XFEL pulse trains that can also be used for other measurements like diffraction methods. The signal to noise of the difference 2D spectrum the authors reported from this measurement is excellent and demonstrates the promise of this approach. Therefore, I generally support this article to be published in the method-oriented journals like Nature Methods. On one hand, I am a bit surprised that such approach has not been practiced yet.

We thank the reviewer for their positive comments. We agree that the concept of the method is rather straightforward, but even those experiments have to be performed, tested, and assessed. Still the implementation is complex and has to be done correctly, and we believe that our account makes this possible. Please find our replies to your questions below.

The data collection time of ~3h with 30 uL/min sample consumption, therefore, about 5mL of total volume, seems to be durable for many biological systems.

Below are more specific comments;

- The SAXS data analysis is complicated and beyond my expertise to evaluate the confidence level of the outcome and needs to rely on the comments of other reviewers, although I agree that the three-phase model fit is better than the two-phase fit in SI Fig. 4. There seems to have an unexplained oscillation in the middle time range of 90 – 225 microseconds in this plot. Would that be another intermediate state?

We thank the reviewer for pointing this out. Indeed, there is a small deviation, but it is close to the scale of the experimental noise so we cannot assign it with confidence to another intermediate state. These small deviations are typical in time-resolved data sets (spectroscopy, scattering, or other) and could in principle be removed by inserting more and more intermediate states; however, overfitting has to be avoided and we believe that it is better to stay with the simplest possible kinetic model. However, future measurements with even better S/N could illuminate this.

- The most uncomfortable part is ~30% elimination of the data that contains artifact. The SAXS difference signal is so subtle and how one can confidently select images with and without artifacts? This was not clearly described in the manuscript.

The jet behavior is sometimes chaotic and some shots will hit its edge or even miss entirely. Those scattering profiles will look very different and we have removed them by comparing the absolute scattering of a run to the mean of the absolute scattering of the same run. This is a standard procedure in the field, however, given that the outlier rate was high in this experiment, we spent a lot of care to evaluate the best strategy to achieve this.

One relevant note is that we are not filtering the data based on the difference signal, but rather the stability of the scattering data over time. We have now inserted text into the SI and the main paper to better describe this.

- What is systematically changing over the data collection time, that makes taking consecutive difference spectra so efficient?

The most likely sources of noise are changes to the GDVN jetting conditions, X-ray beam pointing fluctuations, and the detector noise response. This particular detector is known to have significant systematic noise between different pulses in a pulse train due to the need to handle megahertz intratrain repetition rates. Subtracting consecutive trains helps to mitigate this effect. For the liquid jet, we conducted a more rigorous noise analysis for a separate manuscript (<https://doi.org/10.1107/S2052252523007972>) which is shown below. Plotted here is the integrated scattering normalized by the incoming X-ray fluence for all pulses over a 5 minute run, showing fluctuations of well over 20% from train to train. We believe that detector fluctuations are the main noise sources in the difference scattering signal, but the jet and pointing stability of the X-rays may also contribute.

Reviewer #2:

Remarks to the Author:

The authors describe the use of superconductive EuXFEL to measure protein dynamics with microsecond time resolution.

The authors consider the results presented in this very short, not so detailed paper, a "new implementation".

I find the results intriguing but I don't think that the paper deserve publication in Nature Methods.

Here are some of my concerns:

* flux *

The authors ends up using 175 pulses per train at 10 Hz that is 1750 pulses/s. Assuming 10^{12} ph/pulse, this means $\sim 2 \times 10^{15}$ photons/s

At a 3rd generation synchrotron, 10^9 ph/pulse(10mA) can be delivered to the sample. In few bunch modes, the repetition rate can be as high as few MHz (4 bunches at ESRF are at ~ 1.2 MHz). This translates to $\sim 1 \times 10^{15}$ photons/s very similar to the number obtained at EuXFEL (especially considering high fraction of outliers and the need of 50% duty cycle due to the need of extensive dark measurement).

So from a photon counting perspective the two facilities are quite comparable when looking at microsecond time scale.

We thank the reviewer for the thorough review of this paper.

We agree that the photon counting statistics are important because they ultimately limit the noise levels of the measurement. Recent upgrades at synchrotrons have provided increased flux, collimation and coherence of the X-rays. This has made new experiments possible. However, we believe that in the here-targeted microsecond time range, synchrotrons cannot match high-repetition-rate XFELs.

The numbers regarding the photon flux given by the referee are in our opinion correct. However, their conclusion that the flux at synchrotrons is similar to high-rep rate XFEL is only valid on a "per second" interval. The targeted and biologically interesting time range here is the *microsecond* region and therefore, the comparison should focus on this range.

On the single-pulse level, which is what is required for microsecond time-resolution as in the paper, the XFEL delivers 3 orders of magnitude more fluence (10^{12} versus 10^9 per pulse). Even when sacrificing a significant part of the time resolution, this difference does not change drastically. For example, over $200 \mu\text{s}$ the XFEL delivers 1×10^{14} photons (~ 111 pulses $\times 10^{12}$ photons/pulse), whereas the synchrotron can deliver 2×10^{11} photons (~ 200 pulses $\times 10^9$ photons/pulse). This is a factor of 500. Over an integration range of $500 \mu\text{s}$ the difference is by a factor of 350 and even when extending the time window to 1 ms, the XFEL still outperforms by a factor of 200. Thus, the XFEL will deliver significantly more photons on microsecond timescales.

The reviewer implies that this difference could be overcome by the high pulse rate of synchrotrons, but this requires an area (photon-counting) detector at ~ 1.2 million frames per second. To our knowledge, the fastest detectors currently available at SAXS synchrotron beamlines operate at about 5000 frames per second in continuous mode (23000 fps in burst mode). This means that either the time resolution has to be sacrificed (by integrating over multiple pulses) or the vast majority of the flux is never used to acquire data (because of the detector limitation).

These considerations demonstrate that the microsecond time resolution paired with the unprecedentedly low noise levels that we present in this manuscript are only accessible using high-repetition rate XFEL sources.

Finally, we would like to point out that there is **no published SAXS/WAXS experiment using a trigger-once-read-all mode that reaches the early microsecond time scales.** Work at ID02 at the ESRF, which we consider to be leading in this field, has reached sub-millisecond readout rates of SAXS data, but useful data has only been presented with a 2 ms time resolution <https://journals.iucr.org/j/issues/2023/04/00/uu5008/>. Moreover, we are

not aware of any study, which matches the noise levels (0.001%) of the presented method. We believe that this is due to the fundamental limitations of synchrotrons in terms of X-ray fluence described above: obtaining comparable X-ray fluence is simply not feasible at current synchrotrons/beamlines.

We have included a new figure in the SI to illustrate the difference in "usable" X-ray flux on microsecond timescales between the EuXFEL and synchrotrons, which we hope helps to clarify our rationale. We also added some text in the discussion of the main paper to clarify this.

*** sample consumption ***

The reasons of being able to collect data so effectively, relies a lot on the fast jet ... this implies a huge sample consumption

The authors mentions a flow rate of 30uL/min. Unless, I am mistaken, the final protein concentration is not given (and it should) but I guess this is in the 1 mM range. Running for few (let's say 10, including setup) hours at such flow rate requires the use of 30 uL/min * 60 mins/hour * 10 hours = 18 mL of solution. This is several times higher than available with many (most?) samples.

We thank the reviewer for this comment and we apologize that the amount of sample used was not clearly stated in the original submission. We agree that the sample consumption is an important factor when considering the method's overall viability.

The sample concentration was given in Supporting Information, section "Computation of time-resolved difference X-ray scattering". This value (~1 mM), extrapolated over the 3-hour measurement period, equates to roughly 5.7mL or 75 mg of protein consumed.

Such a quantity, while considerable, is certainly in range for many types of biological samples as was pointed out by Reviewer 1.

In the revised version, we now clearly state the sample amount in the SI (online methods) section. Moreover, we also describe how this consumption could be reduced using segmented jets, which have been developed for XFEL applications:

<https://www.nature.com/articles/s41467-020-18156-7>

As the authors certainly know, most experiments at synchrotrons are performed using few capillaries (with few tens of uL solution) and in some exceptional case using small peristaltic pumps and few ml solutions

In theory, sample volume can be reused for multiple exposures and thus drastically reduce sample use. However, to achieve the low noise levels shown in this paper, the fluence requirements far exceed the radiation damage threshold and prohibit sample reexposure. Consequently, the sample quantity needed at synchrotrons for measurements with equivalent fluence would be on par with an XFEL.

*** detector ***

The authors don't give enough credit to the real workhorse of the experiment they performed, i.e. the AGIPD detector. This (together with the jet) allows to collect data very efficiently.

We fully agree, the fast detector is essential here (see also above). We have emphasized this now more in the last paragraph of the paper.

All in all, I agree that the data quality looks quite nice but I see this as marginal improvement with a very high upfront cost (getting beamtime, producing protein, handling AGIPD data, etc.).

We thank the reviewer for acknowledging the improvement in data quality. We find that the noise level is reduced by a factor of 10, at least, compared to previous synchrotron experiments. This reduction in noise makes it possible to unravel very small signals which we demonstrate. We believe that this opens up for more intricate biological problems to be studied, which is a large step forward.

Maybe the data collection will span over few days instead of three hours, but I am convinced that a well designed synchrotron experiment will result in the same "scientific output" (in terms of understanding) at a much lower cost.

To achieve a detectable fluence of 2×10^{15} photons, as can be collected at the EuXFEL in a second, a synchrotron capable of 10^9 photons/pulses with a 5000 fps detector requires $2 \times 10^{15} / (10^9 \times 5000) = 400$ seconds. So our 3 hours of data collection would turn into 50 days. This probably explains why time-resolved microsecond experiments have not been demonstrated in this way to date.

The paper also sidetrack the attention by discussing the modeling of Light-Oxygen-Voltage (LOV) photosensory domain. I find this section not very useful. I understand the interest (and challenges) of modeling such kinds of data ... but this is just not the right paper. Showing data from more proteins would have been more useful.

We feel that presenting an actual structural result is essential for demonstrating the resolving power of the method. Here, we uncover new details regarding the photocycle of an important photosensory domain protein. This is a prime example of why accessing the microsecond time-range at low noise levels opens up for new discoveries: Alternative methods (NMR, IR spectroscopy, crystallography) have just not been able to address this range even for a much studied sample like AsLOV2. In our view, this further shows the impact of the presented method.

I would be willing to accept a paper along the lines of the one submitted by the authors, only if substantial new information can ONLY be gained using high rep-rate XFELs. Else it might just be another time resolved scattering paper but not a new methodological development.

With the considerations of the *detectable* photon fluence on microsecond timescales given above, and our reply to the remaining comments, we hope that the reviewer can agree that the range of 1-1000 μ s can only be probed with comparable noise levels using high-fluence

MHz XFELs. Synchrotrons should be on par or even better on millisecond time ranges, when many pulses can be integrated without harming time-resolution, however, when focusing on microsecond timescales, the “concentration” of photons offered by high repetition rate XFELs has the potential to unlock new scientific information as we have demonstrated in this manuscript.

Reviewer #3:

Remarks to the Author:

In this paper, Konold et al reported the realization of micro-second time-resolved X-ray scattering using second-generation XFEL and demonstrated this technique in the study of light-initiated J α helix unfolding of a Light-Oxygen-Voltage (LOV) photosensory domain. Many important biological reactions occur at micro-seconds scales, such foldings/unfoldings of protein alpha-helices and DNA/RNA duplexes. However, due to the technical difficulty, there lacks scattering techniques at such time scales. For this reason, this paper could open an avenue to study micro-seconds reactions using x-ray scattering. Given the wide applications of x-ray scattering technique, I believe this paper will attract broad interests among the readership of the Journal. Thus, I would recommend publishing it after the authors address the following questions and comments.

We thank the reviewer for their positive comments on the paper and we have addressed the comments below.

1. Question on the realization of time-resolved x-ray scattering measurements. Considering the high jet velocity (~ 10 m/s), in 300 micro-seconds the solution illuminated by a laser pulse will travel > 3 mm which is much larger than the x-ray beam size (~ 300 nm). The authors mentioned “An alternative approach is to read out a series of probe pulses following a single trigger event. In this way, the efficiency of data collection is vastly improved, reducing sample consumption and suppressing experimental noise through massive averaging”, but failed to disclose the details on how the time-resolution were achieved in their time-resolved x-ray scattering measurements. Was the jet translated for time-resolution or the x-ray beam steered to follow the luminated liquid? If it has been documented previously in any reference paper, please note it is the authors' responsibility to provide sufficient experimental details within the manuscript for the sake of the integrity of the paper.

We thank the reviewer for this question and apologize for the confusion. In our experiment, the bulk of the sample was illuminated within a reservoir inside the transparent 3D-printed GDVN nozzle. Given the slower fluid velocity inside this reservoir, it is possible to excite sufficient volume within the $\sim 325 \times 325 \mu\text{m}$ focal spot, or roughly 1 nL. This excited volume creates a jet that takes 2 ms to move across the interaction region, much longer than the duration of one complete train of X-ray pulses. In this way, fresh sample is provided for each successive probe pulse.

The microsecond time series data then comes directly from exposure of a complete X-ray train with \sim MHz detector readout. **No changes to alignment or timing are required.** The time resolution in this case corresponds to the inverse of the probe repetition rate, or 1.77 μs . The difference data are extracted by subtraction of alternating light and dark trains (5 Hz pump repetition rate)

Careful evaluation of the illumination volume was done to ensure complete sample excitation. These details were explained at the end of paragraphs 2-3, in Fig 1, and also in Supporting Information, section "Optical Excitation Scheme".

We have added further clarification about the illumination conditions in the main text as well as the methods section in SI. Additionally we created the below diagram to hopefully better describe this excitation scheme:

2. Figure 2 and related text.

(a) Fig 2 missed label f.

(b) In Fig2f and text, is Cys adduct the State A? If so, please explicitly label it in the figure and text.

(c) I would suggest labelling Cys450 (and Gln 513), and highlighting the portion of protein that will change, in the ground state structure, to guide the readers eyes.

(a) Label f added

(b) Yes, we have amended as suggested

(c) Upon testing this suggestion, we feel that adding all labels in the dark structure gets a bit crowded and distracting. We would propose adding a small FMN to identify the active site and have done so in the revised manuscript. The main scientific result of the paper relates to the helical changes, which are already labeled and may be colored if necessary. We are also open to suggestions from the reviewer.

3. Comments on the following statements:

" Our data establishes that (i) the Ja helix unfolds in a two-step mechanism within 300 μs , (ii) that it completely unfolds, and (iii) that additional structural changes accompany this process. This concludes a long series of investigations into Ja unfolding[refs 14–19,24], and demonstrates the promising capability of this new time-resolved X-ray scattering method. "

Since the structures used in the data interpretation were picked from a limited conformation space and may not be the only solution, I would suggest weakening the tone on statements (ii) & (iii). For example:

" Our data establishes that the J α helix unfolds in a two-step mechanism within 300 μ s, and also suggests that it completely unfolds and that additional structural changes accompany this process."

We appreciate the reviewer's comments. We agree that these statements need a more toned-down note and will use the suggested sentence instead.

4. Supplementary Figure 5. No $\Delta S_{\text{computed}}$ is in Fig S5a. Is it a typo in (a) caption? In FigS5b, what is $\Delta S_{\text{computed}}$? Difference between theoretical and experimental x-ray scattering for the protein in dark / ground state? Or is it the theoretical difference of x-ray scattering between the ground state and A state?

We thank the reviewer for spotting this mistake and have changed $\Delta S_{\text{computed}}$ to S_{computed} , as the figure legend indicates. In Fig. S5b, $\Delta S_{\text{computed}}$ represents the difference between the computed scattering for a candidate structure of State C and the computed scattering of the ground state structure. This is now clearly defined in the figure caption.

5. Supplementary Figure 6. Since the authors collected the SAXS data on AsLOV2 with different illumination times, I would like to encourage the authors display the full range of SAXS data in Fig S6, because they are closely relevant to the time-resolved measurements, and also it would provide a good comparison between steady state and time-resolved measurements.

We thank the reviewer for their suggestion and have now added an additional panel to Supp Fig 6 showing the full range for the difference scattering recorded as well as updated the figure caption.

Decision Letter, first revision:

Dear Dr. Westenhoff,

Thank you for submitting your revised manuscript "Microsecond time-resolved X-ray scattering by utilizing MHz repetition rate at second-generation XFELs" (NMETH-BC54485A-Z). It has now been seen by the original referees and their comments are below. The reviewers find that the paper has improved in revision, and therefore we'll be happy in principle to publish it in Nature Methods, pending minor revisions to satisfy the referees' final requests and to comply with our editorial and formatting guidelines.

TRANSPARENT PEER REVIEW

Please note: we allow redactions to authors' rebuttal and reviewer comments in the interest of confidentiality. If you are concerned about the release of confidential data, please let us know specifically what information you would like to have removed. Please note that we cannot incorporate redactions for any other reasons. Reviewer names will be published in the peer review files if the reviewer signed the comments to authors, or if reviewers explicitly agree to release their name. For more information, please refer to our FAQ page.

ORCID

Sincerely yours,
Allison

Allison Doerr, Ph.D.
Chief Editor

Nature Methods

Reviewer #1 (Remarks to the Author):

The authors have answered all my questions and concerns in their reply, and I am satisfied with the answers. I don't have further concerns, and I believe, the manuscript is suitable for Nature Methods.

Reviewer #2 (Remarks to the Author):

The authors responses have marginally improved the manuscript. Unfortunately important concerns remain present and overall it is a poorly written methodological paper as discussed below.

**** noise source ****

there is very little discussion on the noise sources. They are mentioned but no effort to understand the origin, it is not even clear what is the expected noise level given the detected number of photons. In the possible noise sources: X-ray wavelength fluctuations/drifts and pointing instabilities should also be considered ... but again just listing all possible noise sources seem to be a bit too simple. A proper discussion would start by calculating the number of photons needed assuming shot noise (Poisson) statistics (as discussed here: <https://www.nature.com/articles/nmeth.1255>); then verifying experimentally how the noise scale with the number of photons to see if (and up to which point) the noise follows the shot-noise. Only at this point, one can claim the the higher number of photons (for microsecond time delays) is an advantage with respect to synchrotrons ... if the dominant noise source is not the photon statistics ... it does not really help much to have some many photons in the first place ...

**** noise level ****

the authors claim "0.001%" but it is not clear how this is calculated. The only place where I can guess a $1e-5$ error bar is in the (time binned) figure 1 but there the total signal is not reported ... and the scales says "cm⁻¹" so I am a bit confused.

Overall the signal to noise does not seem much better than:

- the sub us (!!) dynamics shown in this (10 years old!) paper:

<https://pubs.acs.org/doi/full/10.1021/jp407593j>

- the ~10us dynamics described here: <https://doi.org/10.1038/s42003-018-0242-0>

- the sub-ns dynamics here (2010 paper): www.pnas.org/cgi/doi/10.1073/pnas.1002951107

It is not up to me to do a back-to-back comparison but the overall data quality seems quite similar to the one obtained by the authors ...

The "50 days" the authors mention in the answer to my comment does not take into account the reality ... for time delays that are longer than ~1us, longer x-ray pulses can be used boosting significantly the intensity.

Also faster detectors are being developed (Rigaku 56 kHz XSPA); jungfrau's detector are also more and more spreading; jungfrau has 16 memory cells and the "integration time" can be tuned to the required ~10us (tunable). For each "memory cell frame" ~ $1e11$ photons would reach the sample. I would expect that a limited number of repetitions are necessary to have a very good signal to noise

Clear methods explaining the meaning of the data shown: number of pulses are used, subtraction, azimuthal average, frames rejection, signal normalization, estimation of error bars estimate, how-to-exclude artifacts, etc

In particular, I have the feeling that some low-frequency noise may be contaminating the data, the data in the 0.6-1.4 ang-1 region do not seem very stable ...

**** flux ****

Supp mat 8 is a step into a more useful discussion.

It seems (but no proper description is provided) that the authors "merge" many pulses into relatively large time bins. For the ~tens of microseconds reported in figure 1, synchrotrons can provide much more than the $1e9$ ph/(100 ps pulse). with 10us pulses, about $8e11$ photons can be obtained.

Figure 8 is also misleading since it seems to suggest that synchrotrons can be used only for time scales of the order of $1/\text{frame_rate}$.

A better comparison would discuss in which context (one trigger, many pulses) the calculations are done, would explicitly discuss the needed time resolution (and time range) and how the calculations are done. (why not providing a script in the era of github and such ?)

Figure 8, should also show the delay limit of the proposed approach ($\sim 300\text{us}$) ... this does not apply to synchrotrons.

**** specific points ****

- describe precisely how data have been collected (are different delays averaged in a given range as fig 1 seems to suggest ?)
- publish raw data (at least 1D curves)
- why are the data cut at low q ? what is the minimum q reachable and why ?

Overall, I feel that the paper is methodologically weak. Time resolved scattering to monitor protein changes has been around since ~ 15 years. Tens (or hundreds?) of papers have been published on all sort of systems, time scales, facilities.

A "Nat Methods" paper should provide, in my opinion, a clear step forward in the methodology. This is not the case for this paper.

I hope the authors, and the other reviewers, do not think that I am in a crusade against free electron lasers. I perform and publish several experiments per year using them. I am just worried of over-promising and under-delivering claims that, in my opinion, are starting to hurt the FELs facilities.

All in all, this paper represents a marginal improvement in a very specific time window. With respect to a synchrotron experiment, much more tested and specialized setups, easier detectors and analysis pipeline, much larger window of accessible delays, etc are an overall winner. I would really keep the FELs for those experiments that cannot be done at synchrotrons.

Reviewer #3 (Remarks to the Author):

I think the authors have properly addressed the issues raised in the previous round of review. I recommend publishing this paper as is.

Author Rebuttal, first revision:**Reviewer #1** (Remarks to the Author):

The authors have answered all my questions and concerns in their reply, and I am satisfied with the answers. I don't have further concerns, and I believe, the manuscript is suitable for Nature Methods.

We thank the reviewer and appreciate their recommendation.

Reviewer #3 (Remarks to the Author):

I think the authors have properly addressed the issues raised in the previous round of review. I recommend publishing this paper as is.

We thank the reviewer and appreciate their recommendation.

Reviewer #2 (Remarks to the Author):

The authors responses have marginally improved the manuscript. Unfortunately important concerns remain present and overall it is a poorly written methodological paper as discussed below

** noise source **

there is very little discussion on the noise sources. They are mentioned but no effort to understand the origin, it is not even clear what is the expected noise level given the detected number of photons.

In the possible noise sources: X-ray wavelength fluctuations/drifts and pointing instabilities should also be considered ... but again just listing all possible noise sources seem to be a bit too simple.

A proper discussion would start by calculating the number of photons needed assuming shot noise (Poisson) statistics (as discussed here: <https://www.nature.com/articles/nmeth.1255>); then verifying experimentally how the noise scale with the number of photons to see if (and up to which point) the noise follows the shot-noise. Only at this point, one can claim the the higher number of photons (for microsecond time delays) is an advantage with respect to synchrotrons ... if the dominant noise source is not the photon statistics ... it does not really help much to have some many photons in the first place

** noise level **

the authors claim "0.001%" but it is not clear how this is calculated. The only place where I can guess a $1e-5$ error bar is in the (time binned) figure 1 but there the total signal is not reported ... and the scales says "cm⁻¹" so I am a bit confused.

Overall the signal to noise does not seem much better than:

- the sub us (!!) dynamics shown in this (10 years old!) paper:

<https://pubs.acs.org/doi/full/10.1021/jp407593j>

- the ~10us dynamics described here: <https://doi.org/10.1038/s42003-018-0242-0>

- the sub-ns dynamics here (2010 paper): www.pnas.org/cgi/doi/10.1073/pnas.1002951107

We thank the reviewer for their input on the paper.

Our description of the noise level estimation was indeed not very clear, we have improved on this now. In particular, we clarify the issue with normalization of the data in ED Fig 3, and we also indicate the noise level in the figure as red lines. Additionally, we have now done an estimation of the Poisson noise as the reviewer mentioned, shown in Extended Data Figure 4. Combined, these data indicate that we have a noise level on the order of 10^{-5} .

When it comes to comparison of previously published studies we would like to point out that it can be complicated as there is no "gold standard" for normalization within the field, making data

hard to compare as the noise would vary with the normalization. In order to show something comparable to previous studies, we show selected time points of our difference scattering normalized to the scattering between 1.4-1.6 Å⁻¹ (Extended Data Figure 3, Previously Supplementary Figure 3). From this, together with the Extended Data Figure 4, we conclude that the noise levels are around ~0.001%.

This analysis also demonstrates that most additional noise sources have been eliminated, e.g. jet thickness, varying X-ray intensity and systematic detector errors.

It is not up to me to do a back-to-back comparison but the overall data quality seems quite similar to the one obtained by the authors ...

The "50 days" the authors mention in the answer to my comment does not take into account the reality ... for time delays that are longer than ~1us, longer x-ray pulses can be used boosting significantly the intensity.

Also faster detectors are being developed (Rigaku 56 kHz XSPA); jungfrau's detector are also more and more spreading; jungfrau has 16 memory cells and the "integration time" can be tuned to the required ~10us (tunable). For each "memory cell frame" ~1e11 photons would reach the sample. I would expect that a limited number of repetitions are necessary to have a very good signal to noise

Even if the detector limitation can be removed at synchrotrons (which may be possible in the future, but not presently), the EuXFEL delivers 300 times more photons per q and time bin and we therefore cannot follow the reviewers argument that microsecond time-resolved WAXS using a fast readout detector should reach comparable noise levels to the experiment presented in this paper. Of course, synchrotrons can improve (and probably will) and once faster detectors are widely available, it may be possible to match MHz XFELs, however, until then, we believe that our demonstrated method is unique.

Clear methods explaining the meaning of the data shown: number of pulses are used, subtraction, azimuthal average, frames rejection, signal normalization, estimation of error bars estimate, how-to-exclude artifacts, etc

In particular, I have the feeling that some low-frequency noise may be contaminating the data, the data in the 0.6-1.4 ang-1 region do not seem very stable ...

We thank the reviewer for the input and have clarified the data acquisition strategy in the methods section. There is some noise as in all experimental science, but we did not notice any unnaturally large fluctuations in the mid \AA^{-1} region.

** flux **

Supp mat 8 is a step into a more useful discussion.

It seems (but no proper description is provided) that the authors "merge" many pulses into relatively large time bins. For the ~tens of microseconds reported in figure 1, synchrotrons can provide much more than the $1e9 \text{ ph}/(100 \text{ ps pulse})$. with 10us pulses, about $8e11$ photons can be obtained.

Figure 8 is also misleading since it seems to suggest that synchrotrons can be used only for time scales of the order of $1/\text{frame_rate}$.

A better comparison would discuss in which context (one trigger, many pulses) the calculations are done, would explicitly discuss the needed time resolution (and time range) and how the calculations are done. (why not providing a script in the era of github and such ?)

Figure 8, should also show the delay limit of the proposed approach ($\sim 300\text{us}$) ... this does not apply to synchrotrons.

We are happy to learn that the new figure 8 was appreciated and agree that if multiple pulses are averaged (say for 10us), then the intensity will be increased. This is what the figure shows: Within 10 us integration time at a synchrotron with 10^9 photons/pulse and 1.2Mhz, we would

get approximately 11x the number of photons of a single pulse - which is $1.1 \cdot 10^{10}$ photons. However, this will always be lower than the XFEL situation, where a single pulse delivers 10^{12} photons/pulse. We have included a more extensive description on how the data in ED Fig 8 (now 7 in new count) was computed. We feel that cutting off the data for the XFEL at 300us is not useful, because then the "tailing off" of the XFEL performance over the synchrotron would not be observed in the millisecond time range.

**** specific points ****

- describe precisely how data have been collected (are different delays averaged in a given range as fig 1 seems to suggest ?)
- publish raw data (at least 1D curves)
- why are the data cut at low q ? what is the minimum q reachable and why ?

We thank the reviewer for the comments and inserted a more comprehensive description of the data acquisition. Delays are not averaged, other than for presentation purposes as in figure 1. ED Fig 3 displays data as recorded for single time points. The raw data is deposited into the CXIDB. Lastly, the data is cut at low q due to detector limitations (there is a hole in the detector to let the main beam pass through); in the current setup we were able to collect up to $\sim 0.092 \text{ \AA}^{-1}$

Overall, I feel that the paper is methodologically weak. Time resolved scattering to monitor protein changes has been around since ~ 15 years. Tens (or hundreds?) of papers have been published on all sort of systems, time scales, facilities.

A "Nat Methods" paper should provide, in my opinion, a clear step forward in the methodology. This is not the case for this paper.

I hope the authors, and the other reviewers, do not think that I am in a crusade against free electron lasers. I perform and publish several experiments per year using them. I am just worried of over-promising and under-delivering claims that, in my opinion, are starting to hurt the FELs facilities.

All in all, this paper represents a marginal improvement in a very specific time window. With respect to a synchrotron experiment, much more tested and specialized setups, easier detectors and analysis pipeline, much larger window of accessible delays, etc are an overall winner. I would really keep the FELs for those experiments that cannot be done at synchrotrons.

Here, we disagree with the reviewer: We believe that we have comprehensively shown in document and the rebuttal letters that this method at XFELs does not only surpass current methods in data acquisition rate, but also in S/N. However, the discussion has prompted us to add text to clarify this in the main paper and to discuss the cons and pros of our method compared to synchrotron-based methods, which we feel has improved the paper.

The synchrotron experiments that the reviewer rests their argument on are hypothetical at present. In the future, faster detectors and higher X-ray flux may make similar experiments possible at synchrotrons (however, XFELs will also improve), and until this becomes a reality we believe that our method will provide a means for many researchers to discover protein structural dynamics in the important microsecond time window.

Final Decision Letter:

Dear Sebastian,

I am pleased to inform you that your Brief Communication, "Microsecond time-resolved X-ray scattering by utilizing MHz repetition rate at second-generation XFELs", has now been accepted for publication in Nature Methods. The received and accepted dates will be 20 November 2023 and 10 June 2024. This note is intended to let you know what to expect from us over the next month or so, and to let you know where to address any further questions.

Over the next few weeks, your paper will be copyedited to ensure that it conforms to Nature Methods style. Once your paper is typeset, you will receive an email with a link to choose the appropriate publishing options for your paper and our Author Services team will be in touch regarding any additional information that may be required.

Once proofs are generated, they will be sent to you electronically and you will be asked to send a corrected version within 48 hours. It is extremely important that you let us know now whether you will be difficult to contact over the next month. If this is the case, we ask that you send us the contact information (email, phone and fax) of someone who will be able to check the proofs and deal with any

last-minute problems.

If, when you receive your proof, you cannot meet the deadline, please inform us at rjsproduction@springernature.com immediately.

Please note that *Nature Methods* is a Transformative Journal (TJ). Authors may publish their research with us through the traditional subscription access route or make their paper immediately open access through payment of an article-processing charge (APC). Authors will not be required to make a final decision about access to their article until it has been accepted. Find out more about Transformative Journals

If you are active on Twitter/X, please e-mail me your and your coauthors' handles so that we may tag you when the paper is published.

To assist our authors in disseminating their research to the broader community, our SharedIt initiative provides you with a unique shareable link that will allow anyone (with or without a subscription) to read the published article. Recipients of the link with a subscription will also be able to download and print the PDF. As soon as your article is published, you will receive an automated email with your shareable link.

Please note that you and your coauthors may order reprints and single copies of the issue containing your article through Springer Nature Limited's reprint website, which is located at <http://www.nature.com/reprints/author-reprints.html>. If there are any questions about reprints please send an email to author-reprints@nature.com and someone will assist you.

Best regards,
Allison

Allison Doerr, Ph.D.
Chief Editor
Nature Methods